# High-throughput screen reveals sRNAs regulating crRNA biogenesis by targeting CRISPR leader to repress Rho termination

Ping Lin [1,2], Qinqin Pu[2], Qun Wu [2,3], Chuanmin Zhou[2], Biao Wang[2], Jacob Schettler [2], Zhihan Wang [2], Shugang Qin[2], Pan Gao [2], Rongpeng Li[4], Guoping Li[5], Zhenyu Cheng [6], Lefu Lan [7], Jianxin Jiang[1] & Min Wu [2]

Discovery of CRISPR-Cas systems is one of paramount importance in the field of microbiology. Currently, how CRISPR-Cas systems are finely regulated remains to be defined. Here we use small regulatory RNA (sRNA) library to screen sRNAs targeting type I-F CRISPR-Cas system through proximity ligation by T4 RNA ligase and find 34 sRNAs linking to CRISPR loci. Among 34 sRNAs for potential regulators of CRISPR, sRNA pant463 and PhrS enhance CRISPR loci transcription, while pant391 represses their transcription. We identify PhrS as a regulator of CRISPR-Cas by binding CRISPR leaders to suppress Rho-dependent transcription termination. PhrS-mediated anti-termination facilitates CRISPR locus transcription to generate CRISPR RNA (crRNA) and subsequently promotes CRISPR-Cas adaptive immunity against bacteriophage invasion. Furthermore, this also exists in type I-C/-E CRISPR-Cas, suggesting general regulatory mechanisms in bacteria kingdom. Our findings identify sRNAs as important regulators of CRISPR-Cas, extending roles of sRNAs in controlling bacterial physiology by promoting CRISPR-Cas adaptation priming.

[1] State Key Laboratory of Trauma, Burns and Combined Injury, Institute of Surgery Research, Daping Hospital, Army Medical University, Chongqing, China. [2] Department of Biomedical Sciences, School of Medicine and Health Sciences, University of North Dakota, Grand Forks, ND, USA. [3] Department of Pediatrics, Ruijin Hospital affiliated to Shanghai Jiao Tong University School of Medicine Shanghai, Shanghai, China. [4] Key Laboratory of Biotechnology for Medicinal Plants of Jiangsu Province, Jiangsu Normal University, Xuzhou, Jiangsu, China. [5] Southwestern Medical University, Pulmonary and Allergy Institute, Affiliated Hospital, Luzhou, China. [6] Department of Microbiology and Immunology, Dalhousie University, Halifax, NS, Canada. [7] Shanghai Institute of Materia Medica, Chinese Academy of Sciences, Shanghai, China. Correspondence and requests for materials should be addressed to L.L. (email: llan@simm.ac.cn) or to J.J. (email: hellojjx@126.com) or to M.W. (email: min.wu@und.edu)

CRISPR-Cas systems endow prokaryotes with adaptive and heritable immunity[1–3], which employ RNA-guided nucleases for recognizing and destroying invading DNA or RNA[4–6]. CRISPR-Cas systems function through three phases: spacer acquisition, expression of CRISPR RNA (crRNA), and interference[2,7]. At the spacer acquisition stage, foreign nucleic acids from an invader would be incorporated into CRISPR loci as new spacers, forming expanded archives of past infections. Following the spacer acquisition is the crRNA biogenesis phase, in which CRISPR loci are transcribed to yield precursor crRNA (pre-crRNA) and are then cleaved by Cas proteins into mature crRNAs. In the interference and final phase, crRNA-guided Cas proteins cleave foreign DNA or RNA and mediate their degradation. Although extensive biological and ecological studies have built a framework about the structures and functions of the CRISPR-Cas system[1,8,9], we have just begun to understand the fascinating prokaryote immunity[10]. Importantly, the CRISPR leader sequence exhibits the specificity to constitute an ideal target substrate for spacer integration during the adaptation phase[11], but little is known regarding whether the leader is critical for producing pre-crRNA transcripts during the crRNA biogenesis phase.

Regulatory RNAs are an essential group of molecules that facilitate various aspects in gene expression, such as transcription, RNA processing or stabilization, and translation[12,13]. Small regulatory RNAs (sRNAs), the primary group of regulatory RNA in bacteria (50- to 400 bp), are a major regulator of numerous metabolic and stress responses in bacteria[14–16]. In particular, prompt responses to various stimuli are shown to be controlled by sRNAs[17,18]. However, it is unknown whether sRNAs can modulate CRISPR-Cas adaptive immunity by influencing the transcriptional activity.

Prokaryotes use Rho-dependent termination mechanisms for RNA polymerase (RNAP) recycling in most species of bacteria kingdom[19]. Rho along with its cofactor NusG bind to the transcription terminator pause sites that function as an attenuator[20,21]. Rho moves along the nascent RNA molecules that enable it to function at RNA polymerase, resulting in the dissociation of RNA polymerase complex and termination of transcription.

Here, we used a combination of approaches to search for candidate sRNAs that may regulate CRISPR-Cas function. Based on the data from GRIL-Seq (global small non-coding RNA target identification by ligation/sequencing)[22], the studies derived from genome-wide identification of sRNA in *Pseudomonas aeruginosa*[23,24] and our initial analyses, we constructed high-throughput library encoding 274 sRNAs to ligate to the CRISPR leader via T4 RNA ligase-catalyzed linking assay. The screened candidate sRNAs that target CRISPR leaders were characterized by functional assays and potential binding domains in the targets were predicted using the IntaRNA computing tool and assessed by biochemical assays. PhrS is shown with the most significant interference with Rho-mediated termination by interacting with Type I-F CRISPR leaders, resulting in transcriptional activation of CRISPR loci and then stimulation of CRISPR-Cas adaptive immunity against bacteriophage invasion. As PhrS also shows the similar functions in type I-C/-E CRISPR-Cas systems, demonstrating pervasiveness of sRNA-mediated control of CRISPR-Cas activities. Our data reveal the function of CRISPR leaders, which not only contain a conserved integration host factor to create the ideal target substrate for Cas1-Cas2 to spacer acquisition[11,25,26], but also facilitates crRNA biogenesis by controlling CRISPR loci transcription.

## Results

### sRNA library screening identifies regulators of CRISPR loci.
T4 RNA ligase 1 (single-stranded RNA ligase 1) links two base-paired RNA molecules by catalyzing ATP-dependent formation of a 3′→5′ phosphodiester bond on single-stranded RNAs, which offers a means for investigating the interaction of bacterial sRNAs and RNA molecules in vivo[22]. We used T4 RNA ligase 1 to generate sRNA-RNA chimaera to selectively probe the interactome for interactions between bacterial sRNAs and CRISPR-Cas system (Fig. 1a). We investigated the effect of T4 RNA ligase 1 expression on *P. aeruginosa* PA14 strain throughout the growth period (Fig. 1b), which showed a decline in viability for 1 h after IPTG treatment. Therefore, the inducible expression of T4 RNA ligase 1 was maintained up to 1 h for each experiment.

The *P. aeruginosa* PA14 I-F CRISPR-Cas comprises Cas1, Cas3, Csy1–4 complex flanked by two CRISPR loci (Supplementary Fig. 1a). To identify potential sRNAs that target leaders in CRISPR loci, we used the pKH6 vector[22] to create an arabinose-inducible vector (pKH6-CRISPR1 leader and pKH6-CRISPR2 leader) and introduced the vector into PA14 containing pKH-*t4rnl1*, respectively. After IPTG and arabinose treatment, we used a library of 274 *P. aeruginosa* endogenous sRNAs to detect the ligated chimeric sRNA-CRISPR leader using sRNA-specific primers and CRISPR leader-specific primers as described in Fig. 1a. We observed 9 and 25 sRNA-CRISPR leader chimeras for CRISPR1 and CRISPR2 leaders, respectively (Fig. 1c, d, Supplementary Fig. 1b, and Supplementary data 1). Computational analysis using the online IntaRNA tool also predicts interaction between CRISPR loci and sRNAs (Fig. 1e). The difference between Fig. 1d, e is possibly due to the linking between CRISPR leader and sRNAs through 5′ monophosphates to 3′ hydroxyl groups by T4 RNA ligase 1, but the majority of *P. aeruginosa* sRNA molecules are transcript products containing 5′ triphosphoryl termini. In order to investigate and characterize whether any of these 34 sRNAs interact with and/or regulate CRISPR loci, we constructed each of the sRNA over-expressing plasmids in combination with *CRISPR1-lacZ* or *CRISPR2-lac*Z fusion plasmid, and transformed them into PA14 to monitor lacZ activity. Of the 35 sRNAs tested, one sRNA pant391 repressed CRISPR2-lacZ expression by more than twofold, while sRNAs pant463 and PhrS increased CRISPR2-lacZ expression (Fig. 1f). Of note, PhrS had the strongest positive effect on the level of CRISPR locus, which was further investigated (Fig. 1f).

To detect specific ligation of candidate targets of PhrS with CRISPR loci, we performed reverse transcription-polymerase chain reaction (RT-PCR) to analyze the ligated products as described in Fig. 1a using *phrS*-specific primer and CRISPR locus-specific primer, followed by induction expression of RNA for up to 20 min in the presence of T4 RNA ligase or an inactive T4 RNA ligase (t4K99N). We noticed that the amplicons of PhrS-CRISPR2 leader chimeras were induced to facilitate the expression of PhrS for up to 20 min, but abrogated by an inactive T4 RNA ligase (Fig. 1g). Sequencing analysis of the amplicons confirms that the junction sequences are the PhrS-CRISPR2 leader chimeras (Fig. 1h). These data indicate that PhrS is a candidate sRNA that interacts with type I-F CRISPR leaders of PA14 strain.

### PhrS promotes CRISPR2 locus transcription and interference.
To investigate the influence of PhrS on CRISPR-Cas functionality, we evaluated the expression pattern of *cas* operon or CRISPR loci in the PA14 *phrS* deletion strain (Δ*phrS*) vs. the wild-type (WT) strain. Only CRISPR2 locus, not *cas* operon and CRISPR1 locus, exhibited lower expression in PA14 Δ*phrS* than WT throughout the survey growth period, but restored expression levels close to the WT upon complementing PA14 Δ*phrS* (Fig. 2a). We then measured the transformation efficiency of CRISPR-Cas on eliminating CRISPR-targeted plasmids that contained protospacers in

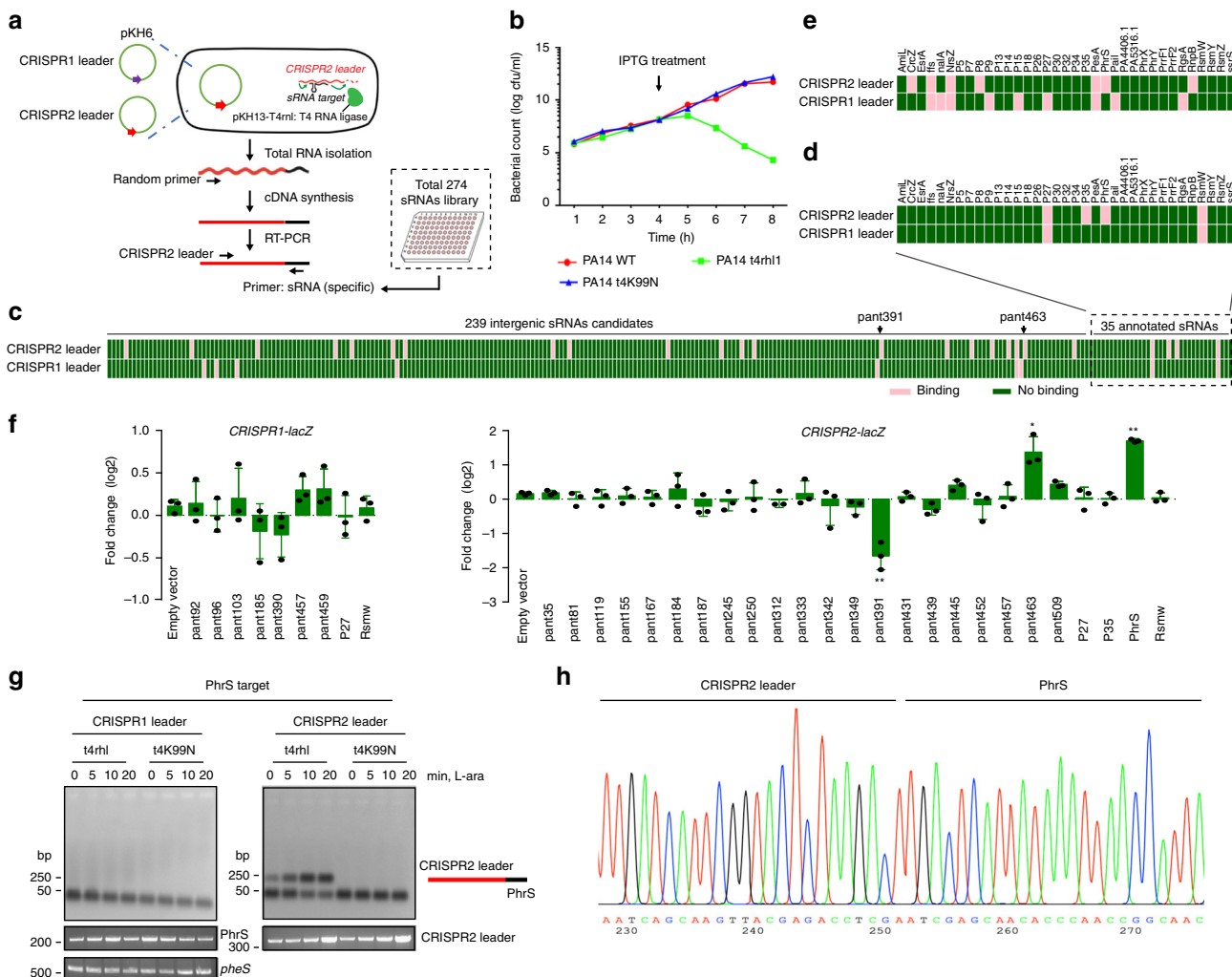

**Fig. 1** T4 RNA ligase-catalyzed ligation of sRNAs to *P. aeruginosa* CRISPR loci. **a** Schematic of the formation of sRNAs chimeras with CRISPR leader by T4 RNA ligase. Two RNA molecules were linked to form pKH6-CRISPR leader plasmid for expressing CRISPR leader and pKH13-*t4rhl1* for expressing T4 RNA ligase. Also shown is reverse transcription-polymerase chain reaction (RT-PCR)-based strategy for determining chimeras of CRISPR leader with sRNA. **b** T4 RNA ligase or its inactive mutation in *t4rnl1* gene with lysine (K) to asparagine (N) affects *P. aeruginosa* cell growth. **c** Screening of 274 *P. aeruginosa* sRNAs library (239 intergenic sRNAs candidate and 35 annotated sRNAs) linking to CRISPR leader by T4 RNA ligase. Pink represents sRNA-containing chimeras; green represents non-target sRNA chimeras. **d** Detection of chimeras of *P. aeruginosa* 35 annotated sRNAs linking to CRISPR leader sequences by T4 RNA ligase in vivo, relative to Supplementary Fig. 1b. Pink represents sRNA-containing chimeras; green represents non-target sRNA chimeras. **e** IntaRNA prediction of *P. aeruginosa* annotated sRNAs interactions with CRISPR leader. **f** Overexpression sRNA to screen candidate sRNAs on regulation of *CRISPR1-lacZ* and *CRISPR2-lacZ* fusion. **g** Amplicons were detected for PhrS-CRISPR2 leader chimeras. Primer for targets PhrS with CRISPR leader (as shown in **a**) was carried out for PCR step. PCR production for PhrS and housekeeping gene (*pheS*) was carried out to ensure the genes expression in all samples. **h** Sequencing reads corresponding to PhrS chimeras with CRISPR2 leader by TA-clone sequencing. Results are shown with mean ± SEM from three independent experiments. **P < 0.01, *P < 0.05, one-way ANOVA plus Tukey test

CRISPR1 (denoted CR1-sp1) or CRISPR2 (denoted CR2-sp1) in PA14 Δ*phrS* (Supplementary Fig. 1a). Strikingly, mutation of *phrS* had no effect on CRISPR1-dependent CRISPR interference (Fig. 2b, left), but resulted in equal transformation frequencies of PA14 ΔTCR lacking *cas* genes when CRISPR2-targeted DNA was used (Fig. 2b, right), reflecting a lack of CRISPR2 interference and immunity functionality that is regulated by PhrS. We also observed that CRISPR-sensitive phage JBD25, which targets a spacer in CRISPR1 locus, failed to replicate in PA14 WT, Δ*phrS* and Δ*phrS*/p-*phrS* (Fig. 2c and Supplementary Fig. 1a). Conversely, CRISPR-sensitive JBD18, which targets a spacer in CRISPR2 locus, was able to replicate in PA14 Δ*phrS*, but failed to replicate in WT and Δ*phrS*/p-*phrS* (Fig. 2c). Taken together, our data demonstrate that PhrS modulates efficiency of CRISPR2 interference, hence controlling its functionality.

Based on these findings, we reasoned that transcriptional changes of CRISPR2 locus may be associated with PhrS. Northern blot analysis supported this premise—that PhrS is required for the synthesis of crRNA in CRISPR2 locus (Fig. 2d). Moreover, addition of CRISPR2 locus into PA14 Δ*phrS* showed sufficient efficiency to account for the CRISPR interference and immunity (Fig. 2e, f), implying that CRISPR2 locus was indeed activated by sRNA PhrS. Meanwhile, expression of CRISPR2 locus in PA14 Δ*phrS* background strain resulted in lowered plaques efficiency of JBD18 (Fig. 2g), corresponding with less potent production of crRNA of CRISPR2 locus in PA14 Δ*phrS*.

Altogether, our results demonstrate that PhrS stimulates CRISPR-Cas-dependent immunity and enhances host defense against invasive element correction by production of the potent crRNA of a specific CRISPR2 locus.

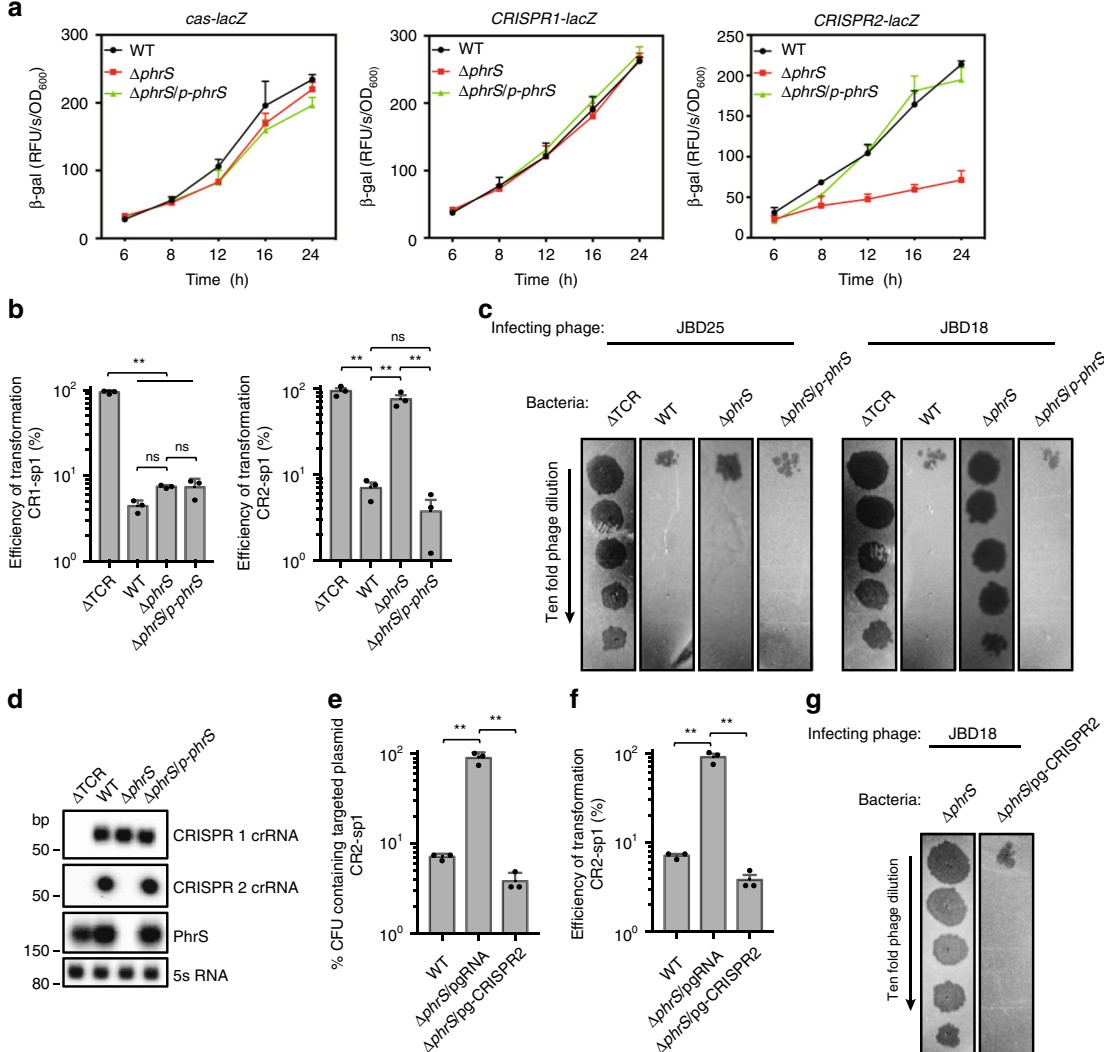

**Fig. 2** PhrS stimulates CRISPR2 crRNA transcription and subsequent CRISPR-Cas interference. **a** cas-lacZ or CRISPR-lacZ activity in PA14 WT and ΔphrS mutant backgrounds throughout the growth period. **b** Transformation efficiency with CR1-sp1 and CR2-sp1 plasmids in PA14 WT or ΔphrS mutant. **c** Phage plaque assay of JBD18 and JBD25 for PA14 WT, ΔphrS, ΔphrS/p-phrS, and PA14 lacking a CRISPR-Cas system (ΔTCR). **d** Northern blot of crRNA levels, PhrS, and 5S RNA in PA14 WT and its mutant strains. **e** Retention of the CRISPR-targeted plasmid CR2-sp1 in the PA14 ΔphrS background strain with pgRNA-CRISPR2 that coexpressed the crRNA in the CRISPR2 locus. **f** Transformation efficiency of CR2-sp1 vector in PA14 ΔphrS background strain with pgRNA-CRISPR2. **g** The same JBD18 phage was tested on PA14 ΔphrS background strain containing pgRNA empty vector or plasmid expression the indicated CRISPR2. Results are shown with mean ± SEM from three independent experiments. **$P < 0.01$, *$P < 0.05$, one-way ANOVA plus Tukey test

**creg motif of PhrS is required for regulating CRISPR system.** In addition to its regulatory function, PhrS has an open reading frames (ORF) that encodes a conserved 37 amino acid peptide (Supplementary Fig. 2a)[27]. We found that there is no difference of CRISPR2 locus transcription between PA14 ΔphrS and ΔphrS/phrS-ORF (restored expression of internal ORF of PhrS) by lacZ reporter and northern blotting (Supplementary Fig. 2b, c) and similarly no difference of CRISPR-Cas interference was noticed (Supplementary Fig. 2d, e). These data demonstrated that the internal ORF of PhrS-encoded protein had no effect on CRISPR-Cas functionality, indicating that PhrS as a sRNA may act on PA14 CRISPR-Cas adaptive immunity.

The secondary structures of PhrS were characterized to contain a conserved region (creg element, 12 nt in length)[28]. To evaluate which motif or region of PhrS is essential for CRISPR-Cas system, we overexpressed three PhrS functional domain variants (pJT-phrS_Δcreg, pJT-phrS_cmut: point mutations introduced into the conserved creg element, and pJT-phrS_creg) in PA14 ΔphrS

background strain (Fig. 3a). As shown in Fig. 3b, pJT-phrS_creg in PA14 ΔphrS background, but not pJT-phrS_Δcreg and pJT-phrS_cmut, stimulated transcription levels of CRISPR2-lacZ fusion gene (as a reporter) similar to that of pJT-phrS. In addition, northern blotting showed that pJT-phrS or pJT-phrS_creg does not cause reduced expression of CRISPR2 locus compared to PA14 WT (Fig. 3c). Furthermore, pJT-phrS_creg showed more efficient transformation inhibition in CRISPR-Cas interference, whereas pJT-phrS_Δcreg and pJT-phrS_cmut did not (Fig. 3d). We also found that CRISPR-sensitive JBD18 can replicate on pJT-phrS_Δcreg and pJT-phrS_cmut, but not on pJT-phrS_creg (Fig. 3e). Our observations suggest that CRISPR2 regulation is dependent on the creg element of PhrS.

As shown in Fig. 1a, we used T4 RNA ligase 1 to link two base-paired RNA molecules. We hypothesized that PhrS acts as a regulatory molecule by interacting directly with a leader sequence to control CRISPR2 locus transcription. Computational analysis by IntaRNA tool showed the potential interaction between PhrS,

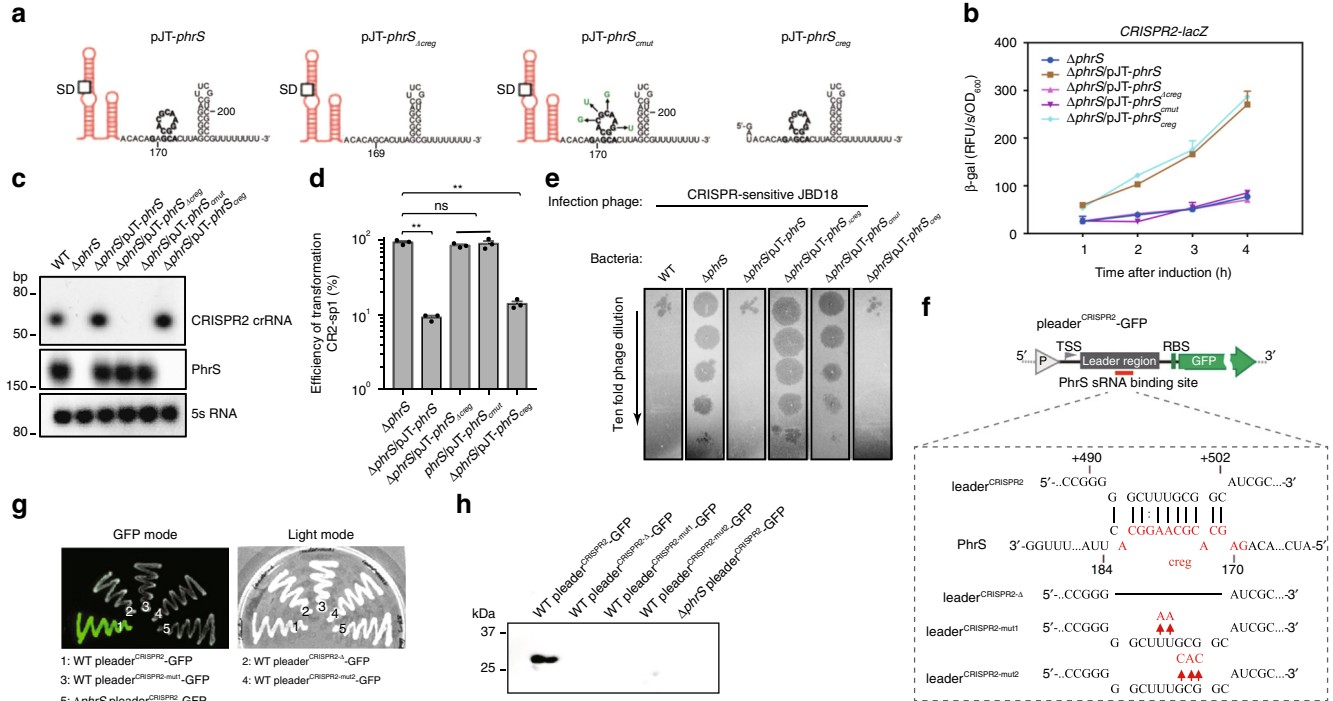

**Fig. 3** creg motif of PhrS is essential for crRNA regulation of CRISPR-Cas adaptive immunity. **a** Diagram of the sRNA PhrS and mutations. creg motif is marked with bold. Eleven nucleotides of creg were deleted in the plasmid pJT-phrS$_{\Delta creg}$. pJT-phrS$_{cmut}$ plasmid contained site mutations in creg motif. pJT-phrS$_{creg}$ represent regions with expression of creg motif. **b** LacZ activity as determined in the PA14 ΔphrS background strain harboring the PhrS expression plasmid pJT-PhrS, plasmid pJT-PhrS$_{\Delta creg}$, plasmid pJT-PhrS$_{cmut}$, or plasmid pJT-phrS$_{creg}$ with toluate (2 mM final concentration). **c** Northern blot of CRISPR2 crRNA production in PA14 ΔphrS background strain containing the PhrS expression plasmid pJT-phrS, plasmid pJT-phrS$_{\Delta creg}$, plasmid pJT-phrS$_{cmut}$, or plasmid pJT-phrS$_{creg}$. **d** Transformation efficiency of CR2-sp1 vector in PA14 ΔphrS background strain within the plasmid pJT-phrS, pJT-phrS$_{\Delta creg}$, pJT-phrS$_{cmut}$, or pJT-phrS$_{creg}$. **e** Phage plaque of CRISPR-sensitive phage JBD18 on PA14 ΔphrS background harboring the plasmid pJT-phrS, pJT-phrS$_{\Delta creg}$, pJT-phrS$_{cmut}$, or pJT-phrS$_{creg}$. **f** IntaRNA (Freiburg RNA tools) prediction of PhrS interactions with CRISPR2 leader target (upper). A diagram of leader binding site with phrS knockout or mutant with red parts mark (lower). **g** GFP fluorescence in PA14 strains or ΔphrS transformed with pleader$^{CRISPR2}$-GFP, pleader$^{CRISPR2-\Delta}$-GFP, pleader$^{CRISPR2-mut1}$-GFP, or pleader$^{CRISPR2-mut2}$-GFP plasmid, respectively. GFP mode is on the left and the visible light of light mode is on the right. **h** Western blot analysis of GFP in the PA14 WT or ΔphrS transformed with pleader$^{CRISPR2}$-GFP, pleader$^{CRISPR2-\Delta}$-GFP, pleader$^{CRISPR2-mut1}$-GFP, or pleader$^{CRISPR2-mut2}$-GFP plasmid, 5 μg total proteins were used to western blotting analysis. Results are shown with mean ± SEM from three independent experiments. **P < 0.01, *P < 0.05, one-way ANOVA plus Tukey test

especially creg element in PhrS, and +491 to +502 segments of CRISPR2 (Fig. 3f, upper), supporting the hypothesis that base-pairing between PhrS and CRISPR2 leader is responsible for PhrS-mediated CRISPR locus transcription. To test this idea, GFP reporter containing three variants of CRISPR2 leader (Fig. 3f, lower) were transformed into PA14 WT. Similar to ΔphrS pleader$^{CRISPR2}$-GFP strain, GFP-containing strain plating analysis demonstrated that mutant-binding sites in the CRISPR2 leader displayed much weaker fluorescence (Fig. 3g). Moreover, western blot analysis of GFP showed that three variants strains had weaker GFP quantity than the WT pleader$^{CRISPR2}$-GFP strain (Fig. 3h). These findings attest that the binding sites in the CRISPR2 locus are highly subject to CRISPR2 locus transcriptional regulation. Taken together, these findings demonstrate that the precise binding sites between creg element of PhrS and CRISPR2 leader is required for the transcription of CRISPR locus to control CRISPR-Cas immunity response.

**PhrS controls Rho-dependent termination at CRISPR2 locus.** We next sought to determine the mechanism how PhrS exerts its function in CRISPR-Cas systems. PhrS as an activator of PqsR synthesis stimulates PQS biosynthesis operon (PqsA-E)[28]. However, mutation of PqsA-E has no effect on CRISPR2 locus transcription (Supplementary Fig. 3a, b) and consequent CRISPR-Cas

interference (Supplementary Fig. 3c, d), indicating that PhrS-mediated PQS biosynthesis operon had no role in CRISPR-Cas expression or function. As some sRNAs modulate gene function through interaction with chromosomal DNAs[14], the "reverse transcription-associated trap (RAT)" assay[29] was performed to detect RNA/DNA interaction through pull-down and PCR by interacting with DNA-specific primers (Supplementary Fig. 4a). We observed no interaction between PhrS and CRISPR2 locus chromosomal DNAs as well as cas1 locus and pheS locus (Supplementary Fig. 4b).

sRNAs are shown to be powerful regulators because they can modulate both transcription and translation[14]. Owing to only pre-crRNA transcribed from CRISPR loci, we focused on PhrS influence on the process of transcription (transcriptional elongation and termination) at CRISPR2 locus. We hypothesized that PhrS may regulate CRISPR2 transcription via inhibition of Rho-dependent transcriptional termination, because there is no G–C rich hairpin loop at CRISPR2 leader for intrinsic termination (Rho-independent termination) to control RNA transcription. To investigate this, we first tested whether Rho terminates CRISPR2 transcription using one round transcription reaction method. Transcription of CRISPR2 template (Supplementary Fig. 5a) gave rise to an intact transcript without Rho (Supplementary Fig. 5b, lane 1). However, Rho together with NusG prompted strong transcription termination (Supplementary Fig. 5b, lane 2), which

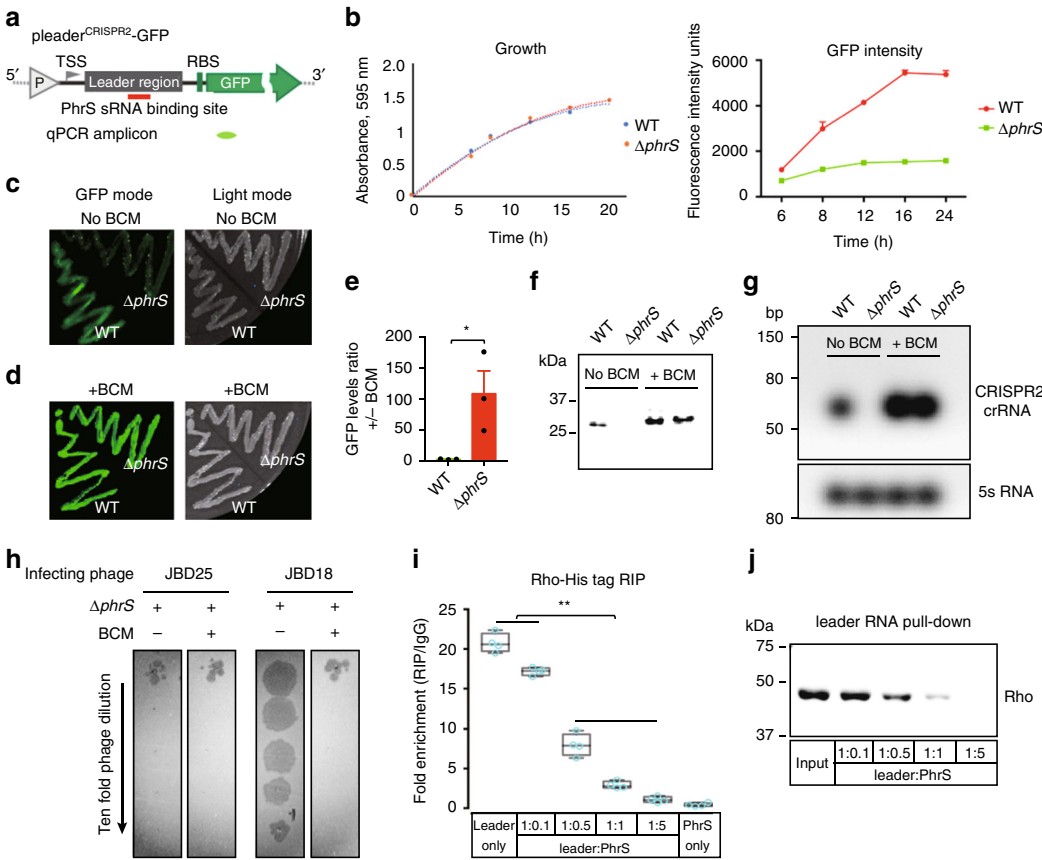

**Fig. 4** PhrS inhibits Rho-dependent termination with leader region within CRISPR2 locus. **a** Diagram of pleader^CRISPR2-GFP construct that includes leader region of CRISPR2 locus. "P" represents Ptac promoter. TSS represents transcription start site. RBS represents a ribosome-binding site. The fragment of qPCR is marked as green. **b** Time-course of GFP levels in PA14 strains (WT and ΔphrS) transformed with pleader^CRISPR2-GFP grew in LB medium. Bacteria density (left) and GFP levels (right) were simultaneously monitored. **c** GFP fluorescence of PA14 WT and ΔphrS harboring pleader^CRISPR2-GFP reporter. **d** GFP fluorescence of PA14 WT and ΔphrS harboring pleader^CRISPR2-GFP cultured on LB with agar containing 8 μg/ml bicyclomycin (BCM). **e** Ratio of *gfp* in PA14 WT or ΔphrS harboring pleader^CRISPR2-GFP without or within BCM. **f** Western blot of GFP in PA14 WT and ΔphrS transformed with pleader^CRISPR2-GFP, 5 μg total proteins were used to western blotting analysis. **g** Northern blot of crRNA levels in PA14 WT and ΔphrS with or without BCM. 5S probe, loading control. **h** Tenfold dilution of lysates of CRISPR-sensitive phages JBD18 and JBD25 grew on bacterial lawns of PA14 WT and ΔphrS without or within BCM. **i** RNA immunoprecipitation (RIP) was performed to test PhrS affecting Rho loading with leader sequence via His-Rho. Box plots: center line represents median, min to max (whiskers) and individual data points (blue). **j** RNA pull-down to investigate PhrS-mediated inhibition of Rho loading with leader region by detection of Rho proteins through western blotting. Bands corresponding to His-Rho were observed in leader RNA pull-down with PhrS in a dose-dependent manner. Results are shown with mean ± SEM from three independent experiments. \*\*$P < 0.01$, \*$P < 0.05$, one-way ANOVA plus Tukey test

was abolished by brief exposure to Rho inhibitor bicyclomycin (BCM)[30] (Supplementary Fig. 5b, lane 3). A GFP reporter with or without the CRISPR2 leader was generated for in vivo experiment (Supplementary Fig. 5c). In comparison with PA14 WT containing pGFP, plating PA14 WT with pleader^CRISPR2-GFP demonstrated significantly subdued fluorescence (Supplementary Fig. 5d), whereas fluorescence of pleader^CRISPR2-GFP strain recovered to the control level in the presence of BCM (Supplementary Fig. 5e). Further, quantitative PCR (qPCR) quantification of *gfp* shows that pleader^CRISPR2-GFP strain exhibited a stronger response to BCM than the pGFP control (Supplementary Fig. 5f, g). In summary, these findings reveal CRISPR2 leader encompassing a Rho-dependent termination signal.

To test whether PhrS influences Rho function on CRISPR2 leader, GFP fluorescence assay in PA14 WT and ΔphrS containing pleader^CRISPR2-GFP plasmid was utilized (Fig. 4a). PA14 WT and ΔphrS containing GFP reporter cultured in luria broth (LB) medium displayed the same growth rate (Fig. 4b, left). Nonetheless, the expression level of GFP gradually increased in PA14 WT rather than the ΔphrS mutant strain throughout the surveyed

growth period (Fig. 4b, right). Considering the Rho activity on CRISPR2 leaders, our data indicate PhrS represses Rho-dependent termination, leading to increased GFP levels. Plating PA14 ΔphrS transformed with pleader^CRISPR2-GFP plasmid displayed markedly reduced fluorescence compared to that of WT (Fig. 4c). Supplementing BCM increased the fluorescence of PA14 ΔphrS (Fig. 4d). Moreover, GFP levels in PA14 WT were altered slightly, while PA14 ΔphrS manifested almost a 100-fold augmentation with BCM treatment confirmed by transcript and protein determination (Fig. 4e, f). This was consistent with the expression of crRNA molecules in the CRISPR2 locus by northern blotting (Fig. 4g). In addition, the plaquing efficiency of phage JBD18 was reduced compared to PA14 ΔphrS with BCM treatment (Fig. 4h). Collectively, these observations support our notion that PhrS inhibits Rho-dependent transcriptional termination to stimulate CRISPR2 crRNA synthesis against phage infection.

To further verify this concept, overexpression of PhrS resulted in the robust increase of *lacZ* transcripts, consistent with the response to BCM (Supplementary Fig. 6a, b). Remarkably, BCM treatment did not lead to comparable induction of CRISPR2

leader in response to PhrS increase (Supplementary Fig. 6c). Altogether, PhrS-mediated repression of Rho in CRISPR2 leader is attributed to the stimulatory effect of PhrS on CRISPR2 transcription.

Rho has to bind RNA to achieve termination[21]. To investigate the mechanism by which PhrS causes the inhibition of Rho, we performed crosslinked RNA immunoprecipitation (RIP) via His-Rho protein to investigate whether PhrS affects Rho loading or translocation with the CRISPR2 leader. We detected the gradually decreased enrichment of the CRISPR2 leader with increasing doses of PhrS (Fig. 4i), indicating that PhrS affects Rho loading. Consequently, to confirm PhrS-mediated inhibition of the biochemical interaction of the CRISPR2 leader with Rho, we performed RNA pull-down assays. The biotin-labeled CRISPR2 leader was transcribed, hybridized to PhrS, and added His-Rho protein for incubation. Samples captured on streptavidin magnetic beads were detected through western blotting analysis to identify and confirm that Rho binding gradually reduced due to the increased PhrS (Fig. 4j). Altogether, PhrS represses Rho loading to stimulate CRISPR2 transcription.

**Reconstitution of PhrS on anti-termination in CRISPR2 locus.** In vitro reconstituted system enabling a single round transcription assay showed that PhrS, similar to BCM treatment, abolished Rho-mediated robust termination (Fig. 5a), meaning that PhrS interacts directly with CRISPR2 leader to inhibit Rho-dependent termination.

Next, we estimated the activity for Rho-mediated termination and PhrS-mediated anti-termination at the CRISPR2 locus. qRT-PCR was performed to determine various regions (UTR [untranslated region], 5′ORF and ORF) of *CRISPR2-lacZ* transcript in PA14 WT and Δ*phrS* strains containing translational CRISPR-lacZ reporter (Fig. 5b). Normalized to "ORF" with each strain, we attained the determination of the termination efficiency within *lacZ* between PA14 WT and Δ*phrS*. Compared to BCM treatment, the [UTR]/[ORF] or [5′UTR]/[ORF] value in PA14 WT without BCM was increased due to powerful Rho-mediated termination (Fig. 5c). Furthermore, a higher value of [UTR]/[ORF] or [5′UTR]/[ORF] in PA14 Δ*phrS* compared to WT inhibited the efficient Rho-dependent termination (Fig. 5c), consistent with a stronger termination phenotype within the CRISPR2 leader in PA14 Δ*phrS* strain. Indeed, treatment of BCM drastically reduced the values of [UTR]/[ORF] or [5′UTR]/[ORF] in PA14 WT and Δ*phrS* (Fig. 5c). To further probe the underlying mechanism, we also detected the native CRISPR2 transcript (pre-crRNA) in PA14 Δ*csy4* and Δ*cys4*/Δ*phrS* strains given Csy4 being responsible for pre-crRNA processing into short crRNAs in PA14[31]. To this end, we found that PA14 Δ*csy4* displayed greatly increased expression levels of pre-crRNAs in Δ*csy4* strains deficient in PhrS (Fig. 5d). [UTR]/[ORF] and [5′UTR]/[ORF] ratio of the transcript of the CRISPR2 locus for PA14 Δ*cys4*/Δ*phrS* strain significantly exceeded PA14 Δ*csy4*, as well as BCM treatment (Fig. 5e). Taken together, these data support the cumulative effect of PhrS on Rho anti-termination as determined in vitro.

**Direct target of PhrS and CRISPR2 inhibits Rho termination.** sRNAs regulate diverse processes through a variety of distinct mechanisms[14]. The precise base-paired sites between the creg element of PhrS and the CRISPR2 leader are required for the CRISPR2 locus transcription (Fig. 3). We hypothesized that PhrS acts as a regulatory molecule by interacting directly with a leader sequence to control Rho-dependent in CRISPR2 locus. To test this notion, we evaluated whether a specific region of the CRISPR2 leader is required, plating PA14 WT was transformed with pleader^CRISPR2(+x+y)-GFP derivatives with various segments of the CRISPR2 leader (Supplementary Fig. 7a). All strains had equal growth rates (Supplementary Fig. 7b, c, lower), but WT pleader^CRISPR2(+400+600)-GFP exhibited much diminished fluorescence, as well as WT pleader^CRISPR2-GFP intensity, compared to other derivative strains (Supplementary Fig. 7b, c, upper), indicating that Rho termination activity may be attributable to a ~ +400 to +600 segment of the CRISPR2 leader, which overlap the binding sites between PhrS and the CRISPR2 leader.

Next, to further test whether the binding sites are required for PhrS-mediated anti-termination in a CRISPR locus, we further generated three variants of the CRISPR2 leader (Figs. 3f and 6a) to perform transcription analysis. We found that anti-termination mediated by PhrS was specific for the binding sites in the CRISPR2 locus, because PhrS has no effect on Rho-dependent termination for other three variants of a CRISPR2 leader region without PhrS-binding sites (Fig. 6a, left). Moreover, Rho together with NusG leaded to obvious transcription termination (Fig. 6a, right, lane 2), which was not suppressed by PhrS with deficiency of PhrS-binding sites in the transcriptional template (Fig. 6a, right, lane 3), but was greatly inhibited by BCM (Fig. 6a, right, lane 4). Furthermore, GFP-containing strain plating analysis also illustrated that the mutant-binding sites of the CRISPR2 leader displayed remarkably attenuated fluorescence (Fig. 6b, upper). Supplementing BCM increased the fluorescence of three variants strains (Fig. 6b, lower). Moreover, qPCR analysis of *gfp* revealed that three variants strains exhibited significantly lower GFP quantity than the WT pleader^CRISPR2-GFP strain (Fig. 6c), consistent with the intensity of the GFP signal (Fig. 6d). GFP expression in the three variants was highly induced after BCM treatment (Fig. 6c, d), indicating that the binding sites of PhrS and CRISPR2 locus are highly subject to the activity of PhrS-mediated anti-termination.

To further substantiate this conclusion, GFP assay was used with PA14 strains: WT, Δ*phrS*, Δ*phrS*/pJT-*phrS_cmut* carrying points mutations introduced into the creg element and Δ*phrS*/pJT-*phrS_cmut*/p-*phrS*, which harbored pleader^CRISPR2-GFP reporter plasmid. Of note, GFP-containing strain plating analysis also demonstrated that mutant creg sites of PhrS in Δ*phrS*/pJT-*phrS_cmut* strain, similar to Δ*phrS* strain, displayed greatly weakened fluorescence, but restored to the control level of the WT upon complementing Δ*phrS*/pJT-*phrS_cmut* strain (Fig. 6e, upper). Supplementing BCM increased the fluorescence of Δ*phrS*/pJT-*phrS_cmut* strain (Fig. 6e, lower). Moreover, qRT-PCR-based quantification and western blot analysis of GFP showed that Δ*phrS*/pJT-*phrS_cmut* strain had much weaker GFP quantity than the WT strain (Fig. 6f, g). Altogether, we conclude that the inhibition of Rho termination is dependent on PhrS direct binding to CRISPR2 leader, resulting in elevated expression of the CRISPR locus in the PA14 I-F CRISPR-Cas system.

**PhrS on anti-termination is a common event for CRISPR system.** To further characterize the role of sRNA PhrS in multiple types CRISPR-Cas regulation, we tested two other types I-C/-E CRISPR-Cas systems, each with at least one CRISPR array, which *P. aeruginosa* ST277 and SM4386 possess PhrS or its homologs with PA14 strains (Supplementary Fig. 8a). We observed that *cas* operon expression, same to CRISPR2 (I-E) and CRISPR1 (I-F) was not altered by PhrS (Supplementary Fig. 8b). Remarkably, CRISPR1 locus-associated type I-C and I-E systems, similar to CRISPR2 (I-F), exhibited significant reduction in PhrS-deficient strains (Supplementary Fig. 8b). In agreement, complementation of PhrS with plasmids restored the expression of CRISPR1 (I-C), CRISPR1 (I-E), and CRISPR2 (I-F).

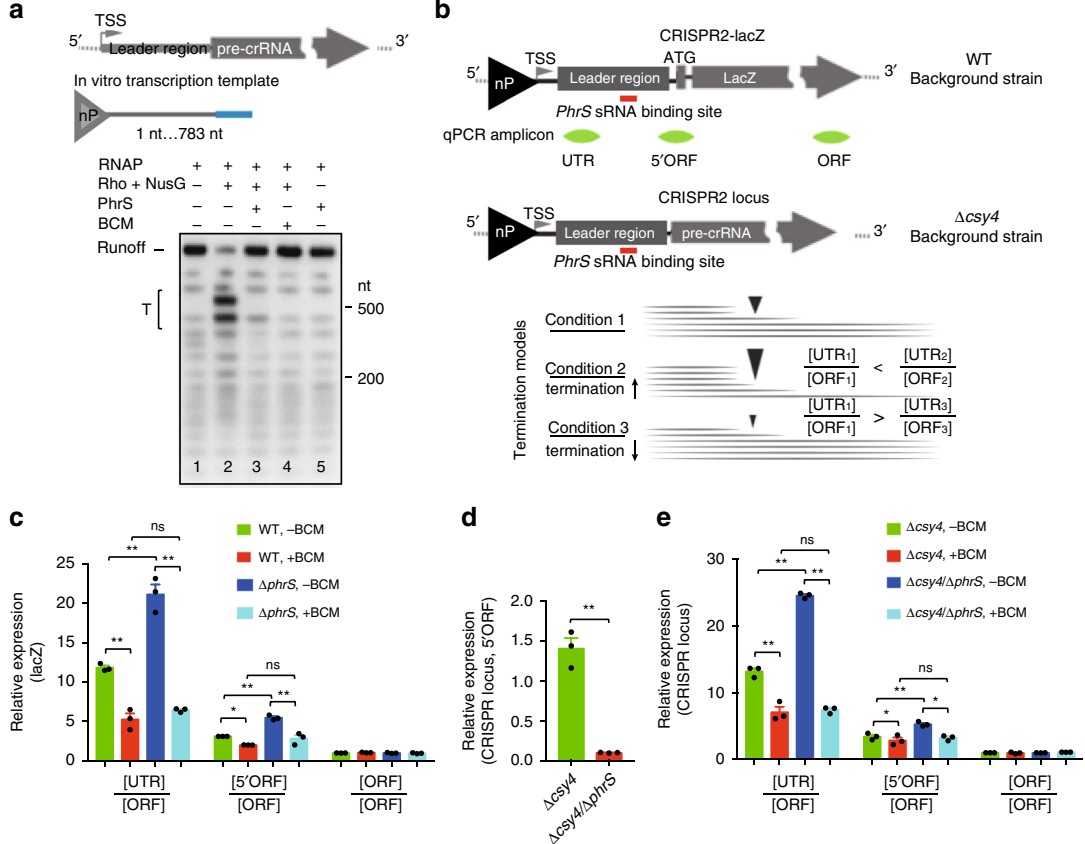

**Fig. 5** Reconstituted PhrS-mediated anti-termination acts on CRISPR2 locus. **a** PhrS sRNA inhibits Rho termination at leader sequence of CRISPR2 locus in vitro. Preformed elongation was transcribed without (lane 1 and 5) or with (lane 2–4) Rho and NusG. The Rho-dependent termination was estimated without (lane 2) or with PhrS sRNA (lane 3) or BCM (lane 4). The bracket represents termination regions (T). **b** Diagram of CRISPR leader with lacZ ORF and CRISPR array. The fragments for qRT-PCR are labeled as green. The dynamic value of [UTR]/[ORF] was used to assess termination efficiency of Rho-dependent termination by PhrS in CRISPR leader, normalized by CRISPR array and *lacZ* RNA. Increased value represents more termination efficiency, but a lower value indicates reduction of termination efficiency. **c** Transcript expression of various *lacZ* zones quantified for PA14 WT or Δ*phrS* before and after BCM treatment. **d** Levels of CRISPR array measured for the PA14 Δ*csy4* and Δ*csy4*/Δ*phrS* mutant. **e** Transcript expression of various CRISPR locus zones quantified for PA14 Δ*csy4* or Δ*csy4*/Δ*phrS* before and after BCM treatment. Results are shown with mean ± SEM from three independent experiments. **P < 0.01, *P < 0.05, one-way ANOVA plus Tukey test

Next, we examined whether PhrS-mediated anti-termination occurs at CRISPR loci in multiple CRISPR-Cas systems. IntaRNA platform analysis showed that these CRISPR loci also possess a potential target site for the creg element of PhrS (Fig. 7a). Monitoring the β-galactosidase activities confirmed that all these CRISPR loci were downregulated in PhrS-deficient strains vs. WT strains (Fig. 7b). Pre-treatment of PhrS-deficient strains with BCM led to an apparent increase of three CRISPR loci (Fig. 7b). Importantly, overexpression of PhrS resulted in upregulation of three CRISPR loci (Fig. 7c). The findings confirm PhrS-mediated anti-termination for type I-C/-E/-F CRISPR loci. Collectively, these observations illustrate that PhrS blocks Rho-dependent termination by targeting CRISPR leaders, which is likely a common phenomenon in multiple CRISPR-Cas systems among the bacteria kingdom (Fig. 7d).

## Discussion

CRISPR-Cas systems are discovered throughout diverse microbes, empowering these microorganisms with unique mechanisms for adaptive immunity[3]. Through a high-throughput approach, we identified sRNA PhrS as a regulator of CRISPR-Cas functionality to stimulate CRISPR loci transcription. sRNAs possess an impressive effect on bacterial behaviors through a variety of

mechanisms, such as changes in RNA conformation[12,18]. Here, we elaborate that PhrS inhibits Rho-dependent termination to ensure CRISPR transcription, establishing a sophisticated principle of sRNA-mediated transcription control of CRISPR-Cas adaptive immunity. Furthermore, the formation of PhrS-leader complex impedes Rho loading on RNA molecules in the CRISPR loci. Our results illuminate a concept that CRISPR leaders not only contain a conserved integration host factor to create the ideal target substrate for Cas1-Cas2 during spacer acquisition[11,25,26], but also facilitate crRNA biogenesis via control of CRISPR loci transcription.

Co-evolutionary dynamics between bacteria and phage presses the emergence of bacterial defense systems[32]. Understanding CRISPR-Cas adaptive immune systems in bacteria has improved our knowledge of bacterial biology and phage–host interaction. While insight into the CRISPR-Cas spacer acquisition and interference stage has expeditiously amassed, the molecular machineries for promptly and precisely stimulating CRISPR-Cas adaptive immunity is rather limited. Previous studies show that bacteria control CRISPR-Cas systems through quorum sensing autoinducers[33,34]. We reveal that PhrS activates the transcription of pre-crRNA in CRISPR-Cas systems. In agreement, PhrS deletion reduces CRISPR-Cas immunity against phage and invading DNA, illustrating an anti-phage mechanism involving

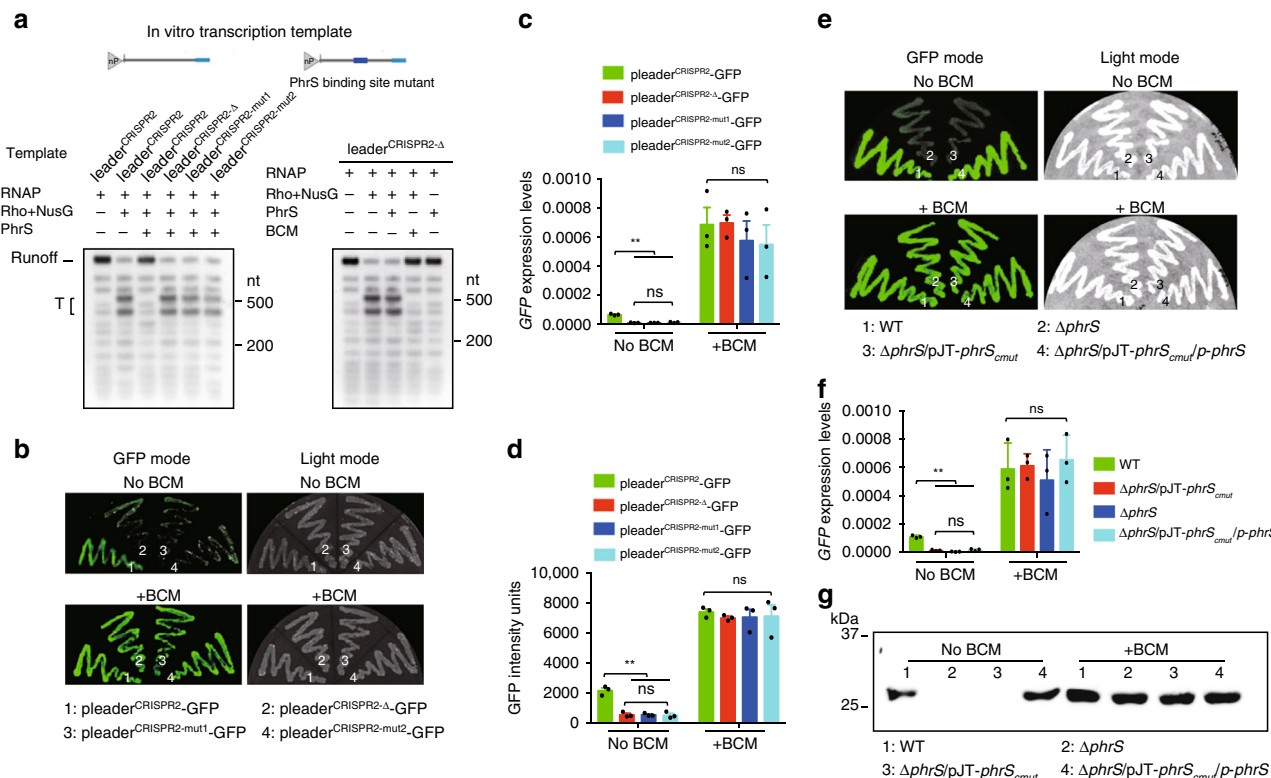

**Fig. 6** The effect of PhrS sRNA on CRISPR transcription via a direct regulatory target. **a** Transcription reaction shows that PhrS did not hamper Rho termination on CRISPR2 leader mutants' template with or without BCM. **b** GFP fluorescence in PA14 strains containing pleader$^{CRISPR}$-GFP, pleader$^{CRISPR-\Delta}$-GFP, pleader$^{CRISPR-mut1}$-GFP, or pleader$^{CRISPR-mut2}$-GFP plasmid, respectively, without (upper) or with (lower) BCM. **c** qPCR for GFP levels with or without BCM for PA14 strains containing pleader$^{CRISPR}$-GFP, pleader$^{CRISPR-\Delta}$-GFP, pleader$^{CRISPR-mut1}$-GFP, or pleader$^{CRISPR-mut2}$-GFP plasmid, respectively. **d** CRISPR2 leader mutation affect GFP intensity. **e** GFP fluorescence in PA14 WT, $\Delta phrS$, $\Delta phrS$/pJT-$phrS_{cmut}$, and $\Delta phrS$/pJT-$phrS_{cmut}$/$p$-$phrS$ strains transformed with pleader$^{CRISPR}$-GFP plasmid without (upper) or with (lower) BCM. **f** qPCR assay for GFP levels without or with BCM for PA14 WT, $\Delta phrS$, $\Delta phrS$/pJT-$phrS_{cmut}$, and $\Delta phrS$/pJT-$phrS_{cmut}$/$p$-$phrS$ strains transformed with pleader$^{CRISPR}$-GFP plasmid. **g** Western blot analysis of GFP in the PA14 WT, $\Delta phrS$, $\Delta phrS$/pJT-$phrS_{cmut}$, and $\Delta phrS$/pJT-$phrS_{cmut}$/$p$-$phrS$ strains transformed with pleader$^{CRISPR}$-GFP plasmid. Ten micrograms total proteins were used in western blotting analysis. Results are shown with mean ± SEM from three independent experiments. **$P < 0.01$, *$P < 0.05$, one-way ANOVA plus Tukey test

PhrS. PhrS stimulates PqsR synthesis to facilitate synthesis of quinolone signal, which links to oxygen availability to impact the formation of *P. aeruginosa* biofilms[28]. Our findings define a regulatory role of sRNA PhrS in CRISPR-Cas activity to battle against phage infection, demonstrating an added layer of regulation in CRISPR-Cas adaptive immunity.

The general termination factor Rho plays an important role in riboswitch-mediated gene regulation that alters the expression of associated protein-coding regions[35,36]. Recently, it was shown that long 5′UTRs of bacterial genes powerfully facilitate Rho-mediated regulatory signals[15]. Here, we provide strong evidence that Rho-dependent transcription termination acts at CRISPR leaders, demonstrating that Rho is required for the control of bacterial non-coding RNA transcription. This phenomenon is also seen in type I-C/-E CRISPR-Cas systems, indicating that CRISPR leader perhaps functions as a general target site for Rho-mediated regulation. Importantly, our findings reveal a surprisingly widespread mechanism in which PhrS mediates anti-termination of transcription by inhibiting the activity of Rho-dependent transcription termination via base-pairing with the regulatory motifs of CRISPR loci leader, which affects the transcription of CRISPR loci. The default repressive state of Rho activity can be partially relieved with the alterations of nutrient or metabolic conditions. For example, Rho is active within several well-characterized *E. coli* riboswitch, such as *thiM*, which responds to intracellular levels of TPP[35,37]. Our data suggest that

the formation of a PhrS-leader complex can inhibit Rho-dependent termination rather than stimulate it. Furthermore, our research leads us to believe that PhrS disrupts Rho loading along with the nascent CRISPR leader or 5′UTR to control transcriptional process in *P. aeruginosa*.

The environmental cues, for instance cell density, may stimulate sRNA expression to modulate bacterial behavior via the base-pairing mechanism[16,38]. Increasing numbers of sRNAs have been characterized, which has significantly improved understanding of their biological function. The regions for base-pairing between sRNA and RNA molecules are marked as seed region. Interestingly, there is only one seed region in some sRNAs, such as RybB[39], whereas numerous seed regions for others matching various RNA molecules, such as Spot 42[40]. The precise mechanism of the creg element of PhrS directly targeted with the CRISPR2 leader is required for CRISPR locus transcription, revealing that the seed regions for specific base-pairing lead to the unwinding of structural elements and ultimately repression of the premature transcription termination in a CRISPR locus.

In summary, our work discovers a mechanism of sRNA-mediated control of CRISPR-Cas systems, showing that PhrS finely regulates anti-termination elements to activate the transcription of CRISPR loci that functions throughout bacterial CRISPR-Cas adaptive immunity in response to phage infection. There are now five major classes of sRNAs in bacteria[18]. Continued identification of diverse classes of sRNAs will expand our

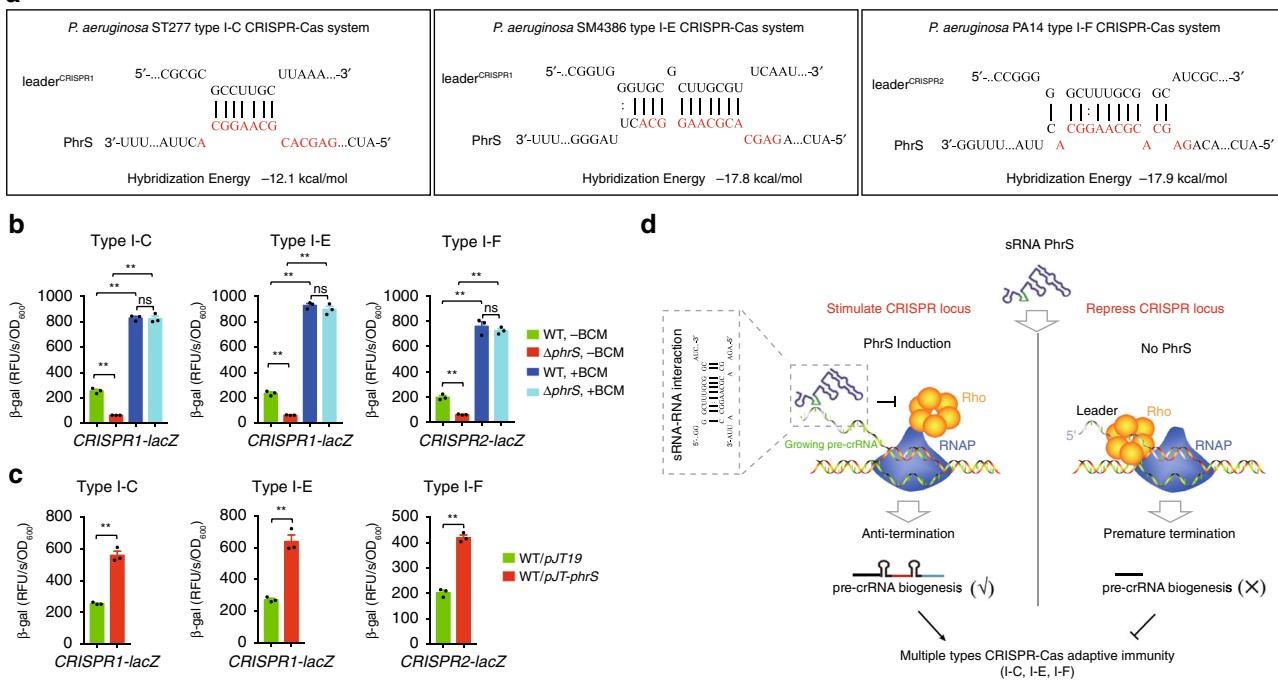

**Fig. 7** PhrS-mediated anti-termination in CRISPR loci of multiple types CRISPR-Cas systems is a general phenomenon. **a** IntaRNA (Freiburg RNA tools) prediction of PhrS sRNA interaction with 5′UTR of CRISPR locus in multiple types CRISPR-Cas system. **b** *CRISPR-lacZ* activity for multiple types CRISPR-Cas system in WT or Δ*phrS* strain with or without BCM. **c** *CRISPR-lacZ* activity for multiple types CRISPR-Cas system in WT without PhrS overexpression, after PhrS overexpression. **d** Proposed model of PhrS-mediated regulation of CRISPR-Cas adaptive immunity. Bacterial small RNA PhrS inhibits the Rho-dependent termination to promote the expression of CRISPR loci, expanding function of sRNA in activating CRISPR-Cas adaptive immunity. Results are shown with mean ± SEM from three independent experiments. **P < 0.01, *P < 0.05, one-way ANOVA plus Tukey test

understanding of their biological action in bacteria, but whether or not other classes of sRNAs or global sRNAs play a pivotal role in regulating CRISPR-Cas systems, especially controlling *cas* operon expression, remains unfolded. In addition, our study reveals sRNA-mediated control of CRISPR-Cas system via RNA–RNA interaction, indicating that other RNA modifications at CRISPR loci, such as epigenetic modifications (M5C or M6A), might also play a part in the transcriptional or post transcriptional control of CRISPR-Cas systems, warranting further investigation.

## Methods

**Bacteria and plasmids**. Bacterial strains (Supplementary data 2) are derivatives of *P. aeruginosa* PA14, ST277 and SM4386 strains. We used *E. coli* DH5a or *P. aeruginosa* strains to construct the plasmids (Supplementary data 2). *E. coli* DH5a or *P. aeruginosa* strains were cultured in LB or LB with agar containing ampicillin (100 μg/ml, Fisher Scientific), chloramphenicol (12.5 μg/ml, Sigma), tetracycline (10 μg/ml, Sigma), kanamycin (100 μg/ml, Fisher Scientific), carbenicillin (100 μg/ ml, Fisher Scientific), gentamicin (75 μg/ml, ACROS) or bicyclomycin (BCM) (8 μg/ml, Santa Cruz).

**Library screen for sRNA chimeras by qPCR**. We used a pKH6 vector from Dr. Stephen Lory of Harvard Medical School[22] to construct pKH6-CRISPR2 and pKH6-CRISPR1 plasmid. These plasmids were each transfected into PA14 containing pKH-t4rnl1, respectively. Overnight cultures of PA14 WT with pKH13-t4rnl1 (or pKH13-t4K99N) and pKH6-CRISPR2 (or pKH6-CRISPR1) plasmid were grown in LB broth with antibiotic and diluted the overnight culture to $OD_{600} = 0.01$. When $OD_{600} = 0.5$, IPTG was added for 1 h incubation and then added the L-arabinose to 20% for different time points. The cells were centrifuged at 12,000 rpm/min for RNA isolation with DNase treatment. One microgram of total RNA was converted to complementary DNA (cDNA) was synthesized with SuperScript III First-Strand Synthesis system (Invitrogen). RT-PCR were performed using GoTaq Green Master Mix (Promega) with specific primers

(Supplementary data 3). The PCR products were recovered and cloned into a pMD-19 vector (Takara) for sequencing.

**Plasmid retention and transformation of efficiency assay**. The plasmid CR1-sp1 and CR2-sp1 were used to perform plasmid retention and transformation of efficiency assay in PA14-WT and its derived mutation strains according to Hoyland-Kroghsbo[33,41]. For plasmid retention assay, PA14-WT and its derived mutants were electroporated with plasmid CR1-sp1 or CR2-sp1 and cultured in LB. Colony forming units (CFU) were calculated on luria broth with agar containing antibiotic to calculate the percentage of plasmid retention. For transformation of efficiency assay, strains transfected with 1 μg CR1-sp1 and CR2-sp1 plasmid were shaken for 1 h at 37 °C and plated on lysogeny broth medium with antibiotic overnight. CFUs were quantified and transformation of efficiency was calculated as the percentage colonies transformed by CR1-sp1 and CR2-sp1 compared with untargeted plasmid.

**Phage isolation and plaque assay**. Phages were isolated from lysogen (Supplementary data 2) according to Marino[42]. Plaque assay on bacterial lawns of PA14 WT or derivatives strains was conducted at 37 °C on LB agar (1.5%) plates with a lower percentage of LBTop agar (0.3%). Added 1 x 10^8 bacteria cells to 4 ml LBTop agar and transformed to LB agar plate as an even layer. The plates were spotted with 3.5 μl of each phage lysate on the lawn and grown overnight. The observed circular zones of clearing indicate the lysis of the tester strains.

**β-Galactosidase assay**. *P. aeruginosa* containing *lacZ* reporters (Supplementary data 2) were grown for β-galactosidase assay for type I-C/-E/-F CRISPR-Cas system according to Joshua P. Ramsay[43]. Briefly, all integrative *lacZ* reporter strains were electroporated and grown in LB with tetracycline at 30 °C to detect β-galactosidase assay according to Adrian G. Patterson[34]. The relative fluorescence intensity was monitored using Bio-TeK Synergy HT Multi-Mode Microplate Reader (Bio-Tek, Winooski, VT). The plate-reader software calculated $V_{max}$ to normalize the value of RFU/s/$OD_{600}$.

**Protein reporter assays**. *Measuring GFP fluorescence*: *P. aeruginosa* PA14 containing GFP reporter were cultured on LB with agar for 20 h, which was imaged at GFP fluorescence mode by IVIS XRII system (PerkinElmer, Waltham, MA).

*GFP assay in the liquid culture for measure GFP intensity*: corresponding strains containing the reporters were cultured to $OD_{600} = 1.0$ and diluted 20-fold to measure GFP in SPECTRAmas GEMINI-XS Spectrofluorometer (Molecular Devices, San Jose, CA).

**RNA isolation and qPCR**. RNA purification with DNAase I digestion was performed by the Direct-zol$^{TM}$ RNA MiniPrep kit (ZYMO RESEARCH, Irvine, CA). cDNA was synthesized using the High Capacity cDNA Reverse Transcription Kit (ThermoFisher Scientific, Waltham, MA). qPCR was analyzed by Maxima SYBR Green qPCR Master Mix (ThermoFisher Scientific, Waltham, MA) with gene-specific primers (Supplementary table 1).

**Northern blot analysis**. Five microgram of total RNA was separated by 6% TBE-urea polyacrylamide gels in 1x TBE and then transferred to Hybond-XL membranes (GE-Healthcare, Pittsburgh, PA). Using ultraviolet to cross-linking and hybridizing with gamma$^{32}$P-ATP labeled oligonucleotide probes (Supplementary table 1). After washing three times to remove unwanted probe, the bands were detected.

**sRNA PhrS synthesis**. PhrS templates with T7 promoter sequence (Supplementary table 2) were amplified by RT-PCR. PhrS sRNA was transcribed with MEGAscript T7 kit according to manufacturer's protocols with TURBO DNase treatment.

**In vitro transcription**. For transcription reaction, the procedure was previously described by Sedlyarova et al.[15] with modification. Briefly, the initial elongation complex was formed with 75 nM of corresponding transcription templates (Supplementary Table 2) and 100 nM RNA polymerase (RNAP) holoenzyme in 100 µl of 40 mM Tris-HCl pH = 8.0; 20 mM MgCl$_2$; 50 mM NaCl; 0.003% IGEPAL; 5 mM β-mercaptoethanol with 2 µl RNase inhibitor (Takara Bio USA, Mountain View, CA). Transcription reaction mix (25 µM GTP, UTP, ATP-P-32) was performed at 22 °C for 5 min incubation and then mixed with 1 µM Rho with NusG proteins with or without the PhrS sRNA or bicyclomycin. Heat the mixture containing 1 mM ATP and 100 µM other NTPs to transcribe at 37 °C with 10 min and stop the reaction with 1x TBE, 8 M Urea, 20 mM EDTA, 0.025% xylencyanol, 0.025% bromophenol blue at 95 °C. The samples were separated by 6% TBE-urea polyacrylamide gels.

**Western blotting**. The samples derived from PA14 WT and its derived mutation strains were separated and transferred to nitrocellulose membranes (GE-Healthcare, Pittsburgh, PA). Membranes were incubated with mouse monoclonal antibody against GFP (Biolegend, San Diego, CA) and His-tag (ThermoFisher Scientific, Waltham, MA) at 1:5000 for overnight at 4 °C with primary antibodies[44,45]. After washing, adding corresponding secondary antibodies for 1.5 h. After washing five times with washing buffer, the protein bands were visualized by chemiluminescence.

**RNA immunoprecipitation (RIP)**. Anti-His antibody was used for RIP assay. One microgram of His-Rho were used to pull down the RNA of CRISPR leader incubated with different concentration of PhrS. The reactants were washed three times with RIPA buffer (150 mM NaCl, 50 mM Tris, 0.5% sodium deoxycholate, 0.1% SDS, 1% NP-40 with RNase inhibitor [Takara Bio USA, Mountain View, CA]) and then washed twice with another washing buffer (1 M NaCl, 50 mM Tris, 0.5% sodium deoxycholate, 0.1% SDS, 1% NP-40). RNA isolation was used Direct-zol$^{TM}$ RNA MiniPrep kit (ZYMO RESEARCH, Irvine, CA).

**Biotinylated RNA pull-down**. We used the Biotin RNA labeling mix (Sigma) and T7 RNA polymerase to get the Biotin-CRISPR leader transcripts. Biotin-CRISPR leader incubated with different concentration of PhrS were adsorbed onto streptavidin magnetic beads and added His-Rho protein to incubation at 4 °C for 6 h. After washing five times in RIP buffer (150 mM KCl, 25 mM Tris-HCl (pH 7.4), 0.5 mM DTT. 0.5% NP-40, and protease inhibitors), the eluted samples were detected by western blot.

**Statistical analysis**. Values were obtained from three independent experiments, shown as mean ± SEM. *P*-values were derived with GraphPad (GraphPad Software, LaJolla, CA) using one-way analysis of variance (ANOVA) plus Tukey test.

**Reporting summary**. Further information on research design is available in the Nature Research Reporting Summary linked to this article.

## Data availability
Authors confirm that all data and materials in the study can be obtained from the corresponding author (M.W.) upon reasonable request.

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

## Acknowledgements

We thank Dr. Stephen Lory of Harvard Medical School to give pKH6, pKH13, or lacZ vectors, Dr. Evgeny Nudler of New York University School of Medicine for providing Rho and NusG proteins, Dr. Elisabeth Sonnleitner of University of Vienna for providing pJT-phrS, pJT-phrS$_{\Delta creg}$, pJT-phrS$_{creg}$, and pJT-phrS$_{cmut}$, Dr. Alan R. Davidson of University of Toronto for providing *Pseudomonas aeruginosa* SMC4386, Dr. Shiyun Chen of Wuhan institute of Virology (Chinese Academy of Science) for providing information of sRNA library information, and Dr. Sergei Nechaev and Dr. Catherine Brissette of University of North Dakota for critical reading. This work was supported by National Institutes of Health Grants AI101973-01, R01AI109317-01A1, and R01AI0138203-01 to M.W., UND Post-Doc Pilot Grant; this work was also supported by the Key Program of National Nature Science Foundation of China (81530063) to Jianxin Jiang. The funders had no role in study design, data collection and analysis, decision to publish, or preparation of the manuscript.

## Author contributions

P.L., M.W. and J.X.J designed experiments and wrote the manuscript. P.L., Q.P., Q.W. and J.S. designed and performed most of the experiments. P.L., M.W. and J.X.J. analyzed data. C.Z., B.W., S.Q., R.L., P.G., G.L., Z.W., Z.C. and L.L. advised on experimental design and manuscript preparation.

## Additional information

**Competing interests:** The authors declare no competing interests.

