## [Peer Review File · Nature Communications]

Reviewers' comments:

Reviewer #1, expert in bacterial sRNA (Remarks to the Author):

In this study, Lin et al. set out to test whether small, regulatory RNAs (sRNAs) in *Pseudomonas* impact the CRISPR loci. To this end, they tested whether any of the sRNAs found in a library encoding 274 of these regulators ligate to the CRISPR2 or CRISPR1 leader when T4 RNA ligase is expressed in the cell (a variation of GRIL-seq developed by the Lory lab). After carrying out RT-PCR to detect chimeras, the authors focused on the PhrS sRNA, which they report regulates the CRISPR2 locus via Rho-dependent transcriptional termination. While this is an intriguing model, the study should be improved in a number of ways:

1. The manuscript should be restructured to document base pairing between PhrS and the CRISPR2 leader before examining the effect on transcription termination (i.e. have the following order: (a) GRIL-seq to identify sRNAs which bind CRISPR leaders, leading to focus on PhrS, (b) impact of Δ phrS on CRISPR2, (c) effect of deletions and compensatory mutations, (d) regulation via Rho.)

2. The authors should refrain from using the term "bind" in the absence of data to support base pairing. Ligation resulting in chimeras only infers proximity. Thus, the authors should use the nomenclature sRNA-CRISPR leader chimeras. Additionally, programs such as IntaRNA (page 232) only predict binding. The data in Figure 1e (Line 80) do not strengthen the identification of sRNAs which potentially bind the CRISPR leader sequences. IntaRNA would be better served as a way to confirm base-pairing regions within the positive hits identified through GRIL-seq.

3. The authors need to provide more support for a physiological interaction between PhrS and the CRISPR2 leader by:

--Documenting that PhrS-CRISPR2 chimeras are observed with chromosomally expressed PhrS and CRISPR2.

--Probing the northern in Figure 2d for PhrS.

--Probing the northern in Figure 3c for PhrS.

--Generating and assaying compensatory mutants in PhrS to test whether they restore base pairing and regulation.

The authors should also discuss the physiological relevance of PhrS mediated regulation in terms of what is already known about PhrS.

4. The authors need to carry out additional experiments to elucidate the mechanism by which PhrS causes antitermination (by inhibiting Rho loading or translocation or by yet another mechanism).

5. The data presented in this study do not "reveal that bacterial intracellular signaling modifies the sequences of RNA molecules..." as stated on Page 11.

6. Figures need to be improved or commented on as follows:

--General: The choice of colors in the figures is distracting and confusing. There are too many different colors and some do not match the descriptions.

--Figure 1c and 1d: Pant391 and Pant463 should be labeled.

--Figure 1f: All bars should be labeled. It is unclear why these specific sRNAs were tested with the lacZ constructs. The authors should test all the positive hits (identified by GRIL-seq) with the lacZ clones, together with a few negative controls.

--Figure 1g and Lines 88-89: It is not clear what is shown in this figure. What is meant by "unique amplicons"?

--Figure 2c and Lines 102-104: This image is of very poor quality and should be improved. The authors should describe the phage JBD25 and JBD18 and their differences in more detail in the text.

--Figure 2d: The authors should provide an explanation for why CRISPR1 crRNA levels increase in Δ phrS.

--Figure 3a: The authors need to disclose the evidence for the secondary structure of PhrS.

--Figure 3c: The authors should note that the levels of 5S are significantly lower in the third lane and/or repeat this northern.

7. Editorial comments:

--Throughout: The manuscript needs to be checked for typographical errors such as Page 4, Line 111 "Consistent with these, expression...", Page 6, Line 160 "Likewise, plating assay showed...", Page 9, Line 283 "The precise mechanism of crew element of PhrS via direct..."

--Throughout: sRNAs are typically written with the first and last letter capitalized and no italics i.e. "PhrS"

--Line 162: The authors need to define the "creg" and "cmut" sequences of PhrS. What does this nomenclature mean? Is this a known motif? Why was this region chosen for mutational analysis? It would be helpful to show the base-pairing region on CRISPR2 at this point in the text to drive the reasoning for mutating the creg sequence.

Reviewer #3, CRISPR expert (Remarks to the Author):

Lin et al uncover a novel small RNA-mediated mechanism by which Type I CRISPR-Cas systems are regulated in *P. aeruginosa*. The small RNA PhrS is the main focus of the paper—to identify PhrS, the authors first used a screen that relies upon T4 RNA ligase to covalently ligate any small RNAs that might be bound (via base-pair interactions) with the CRISPR1 and 2 leader regions. RT-PCR was then used to amplify sRNA-CRISPR leader hybrids and identify the small RNA linked to the leader. Following this screen, PhrS was identified along with a few other sRNA candidates. The authors then used genetic analyses to convincingly demonstrate that PhrS promotes efficient expression of the CRISPR 2 locus (but not the CRISPR 1 locus) and is required for CRISPR2 interference against plasmid transformation and phage infection. Using *in silico* analyses coupled with site-directed mutagenesis, the critical region in PhrS required for its function was identified (a so-called creg motif), and the region in the CRISPR leader that likely binds this small RNA was also found. Next, a series of *in vitro* and *in vivo* experiments were conducted to demonstrate that PhrS promotes transcription of the CRISPR locus by helping RNA polymerase to bypass a rho-dependent termination signal that appears in between the promoter and the first direct repeat in the CRISPR locus. Finally, this mode of regulation was shown to be present in other Type I CRISPR-Cas systems found in other strains of *Pseudomonas*.

Overall, this study constitutes a substantial amount of work that convincingly demonstrates the precise mechanism by which PhrS regulates the function of Type I CRISPR-Cas systems in *P. aeruginosa*. The flow of logic is very clear, and the authors have anticipated and answered many of my questions. I expect that this work will be well-received by the CRISPR community because little is known about how CRISPR systems are regulated, and to my knowledge, this constitutes the very first report of a small RNA-mediated mechanism of CRISPR regulation. I have minor suggestions, most of which will help improve clarity and readability. They are as follows in no particular order:

1. To maintain high standards of rigor and reproducibility, it would be important to note the number of replicates that were performed for each experiment, either in the figure legends or materials and methods section.
2. Early in the results section (around lines 74-75), there should be a brief description of the architecture of the CRISPR loci in *P. aeruginosa* and a reference to Figure S1A.
3. The difference between Figure 1d and 1e is very striking--there is almost no overlap with predicted interacting sRNAs and actual ones detected by the screen. Given that this screen can only detect interactions with sRNAs that have a 5'-monophosphate, while the majority of bacterial sRNAs have 5'-triphosphates, it would be important to indicate that the screen was in fact biased, and remove the word "unbiased" from the Discussion (line 243).
4. Somewhere around lines 102-105, it is important to explain in the text which CRISPR locus has a spacer that targets each phage, and refer to Fig. S1A. Without getting this information in the main text, it is impossible for the reader to interpret the phage plating data in Figure 2.
5. In lines 118-120, you should explain how you excluded possibility of the internal ORF having an effect. I know that the data appears in the supplementary figures but a brief explanation of this data should be included in the text.
6. As above, lines 133-136 rule out some possible mechanisms of action of PhrS and simply refer

the reader to Figures S3 and S4. It would be important to give a brief explanation of what the data is showing so that the reader can follow.

7. Regarding Figure S4, what is the positive control? Also, it seems strange that one step of the procedure involves "nuclei collection" when bacteria do not have nuclei. Please revise the description of the protocol to be more accurate.

8. Line 135—please change "chromatin" to "chromosomal" since bacteria do not have chromatin.

9. Lines 155-156—please fix the grammar in that sentence.

10. Line 223—what is meant by "constituted"? Please use a better descriptive word.

11. Line 224—Please change "another" to "other".

12. Lines 293-297—The data does not appear to support the claim that the data shows "...bacterial intracellular signaling modifies the sequences of RNA molecules to control CRISPR-Cas systems,..." Please revise to more accurately reflect the findings.

13. In Fig. 1F, please label the small RNAs that each bar corresponds to. Also, Pant 463 and 391 do not appear to be in the annotated list. Are they among the non-annotated ones?

14. In Fig. 5A, please change "pro-crRNA" to "pre-crRNA".

15. In Fig. 6B, please change "Phrs" to "PhrS".

Point-by-point responses to Referees' comments:

Dear reviewers,

First, I would like to express my heartfelt appreciation to both of you for recognizing the significance of our study and for evaluating our work and giving the important suggestions to improve our work. We have carefully revised the manuscripts according to these invaluable comments. We have highlighted the changes in the text (yellow) of the manuscript and made point-by-point responses to each of the reviewers' comments below.

Reviewers' comments:

Reviewer #1, expert in bacterial sRNA (Remarks to the Author):

In this study, Lin et al. set out to test whether small, regulatory RNAs (sRNAs) in Pseudomonas impact the CRISPR loci. To this end, they tested whether any of the sRNAs found in a library encoding 274 of these regulators ligate to the CRISPR2 or CRISPR1 leader when T4 RNA ligase is expressed in the cell (a variation of GRIL-seq developed by the Lory lab). After carrying out RT-PCR to detect chimeras, the authors focused on the PhrS sRNA, which they report regulates the CRISPR2 locus via Rho-dependent transcriptional termination. While this is an intriguing model, the study should be improved in a number of ways:

1. The manuscript should be restructured to document base pairing between PhrS and the CRISPR2 leader before examining the effect on transcription termination (i.e. have the following order: (a) GRIL-seq to identify sRNAs which bind CRISPR leaders, leading to focus on PhrS, (b) impact of $\Delta phrS$ on CRISPR2, (c) effect of deletions and compensatory mutations, (d) regulation via Rho.)

Response: We highly appreciate the careful reading and the great suggestion from reviewer 1. We have now re-organized the manuscript exactly as suggested and added the information about the base pairing between PhrS and the CRISPR2 leader as a necessity for CRISPR2 regulation before examining the effect on CRISPR transcription by Rho-dependent termination (line: 151-163). We believe that the reorganization has significantly improved this manuscript and made it more logical, coherent and easy to follow.

2. The authors should refrain from using the term “bind” in the absence of data to support base pairing.

Ligation resulting in chimeras only infers proximity. Thus, the authors should use the nomenclature sRNA-CRISPR leader chimeras.

Response: We apologize for using the term “bind” in the occasions without direct binding assays to support base pairing specific binding. We have carefully proofread the entire manuscript several times to correct all errors and changed “bind” into “predicted interaction” and also the nomenclature sRNA-CRISPR leader chimeras, such as line 24, line 86-87, line 564, line 572-573, line 575-576.

-Additionally, programs such as IntaRNA (page 232) only predict binding. The data in Figure 1e (Line 80) do not strengthen the identification of sRNAs which potentially bind the CRISPR leader sequences. IntaRNA would be better served as a way to confirm base-pairing regions within the positive hits identified through GRIL-seq.

Response: We extremely appreciate your careful reading and great comments. We have rewritten this section and the entire manuscript to improve the overall quality and presentation of our study. For example, we revised (i) “Computational analysis using the online IntaRNA tool also suggests that CRISPR loci have binding sites interacting with sRNAs” to “Computational analysis using the online IntaRNA tool also predicts interaction between CRISPR loci and sRNAs” (line 88-89); (ii) “Computational analysis by IntaRNA tool showed the interaction between PhrS, especially creg element in PhrS, and +491 to +502 segments of CRISPR2” to “Computational analysis by IntaRNA tool showed the potential interaction between PhrS, especially creg element in PhrS, and +491 to +502 segments of CRISPR2” (line 153-154); (iii) “IntaRNA platform analysis showed that these CRISPR loci also possess a binding site for creg element of PhrS” to “IntaRNA platform analysis showed that these CRISPR loci also possess a candidate target site for creg element of PhrS” (line 288-289).

GRIL-seq is shown to be valuable for screening global small non-coding RNA targets by ligation and sequencing to identify the chimeras¹. Based on the existing GRIL-Seq data¹, other genome-wide studies of sRNA^{2,3}, and our early observations, we have constructed a library to identify potential sRNA regulators for CRISPR-Cas systems and successfully identified and validated PhrS as a novel regulator of CRISPR-Cas adaptive immunity (line 57-63).

3. The authors need to provide more support for a physiological interaction between PhrS and the CRISPR2 leader by:

--Documenting that PhrS-CRISPR2 chimeras are observed with chromosomally expressed PhrS and CRISPR2.

Response: Thanks for the wonderful suggestion. We have now performed additional experiments to observe with chromosomally expression as well as a housekeeping gene (*pheS*) expression as an internal control (Fig. 1g), which support our conclusion that PhrS-CRISPR2 chimeras were observed.

--Probing the northern in Figure 2d for PhrS.

Response: Thank you very much for your valuable opinions. We have performed additional experiments to probe the northern blot for PhrS (Fig. 2d). We found that PhrS expressed in the PA14 WT and $\Delta phrS/p-phrS$, but not in the $\Delta phrS$.

--Probing the northern in Figure 3c for PhrS.

Response: Thank you very much for your valuable opinions. See detail results in Fig. 3c for newly added northern blotting analysis. We have revised according to the reviewer's advice.

--Generating and assaying compensatory mutants in PhrS to test whether they restore base pairing and regulation.

Response: Thanks so much for the constructive suggestions. We have now performed the additional experiment to investigate the restoration effects. We generated the mutation of the creg motif in PhrS (Fig. 3a) and compensatory mutants. In brief, we performed a GFP assay similar to that described above in PA14 strains: WT, $\Delta phrS$, $\Delta phrS/pJT-phrS_{cmut}$ carrying points mutations introduced into creg element, and $\Delta phrS/pJT-phrS_{cmut}/p-phrS$. These strains were transformed with pleader^{CRISPR2}-GFP transcriptional fusion plasmid. Of note, GFP-containing strain plating analysis also demonstrated that mutant creg sites of PhrS in $\Delta phrS/pJT-phrS_{cmut}$ strain, similar to $\Delta phrS$ strain, displayed greatly weakened fluorescence, but restored to the control level of WT upon complementing $\Delta phrS/pJT-phrS_{cmut}$ strain (Fig. 6e, upper). Supplementing BCM increased the fluorescence of $\Delta phrS/pJT-phrS_{cmut}$ strain (Fig. 6e, lower). Moreover, qRT-PCR-based quantification and western blot analysis of GFP showed that $\Delta phrS/pJT-phrS_{cmut}$ strain had much weaker GFP quantity than the WT strain (Fig. 6f,g), demonstrating that precise binding sites lying between creg element of PhrS and CRISPR2 leader are required for CRISPR locus transcription (line 266-276).

The authors should also discuss the physiological relevance of PhrS mediated regulation in terms of what is already known about PhrS.

Response: Thank you for this excellent suggestion. In the carefully revised manuscript, we have added the content about previous research regarding PhrS, such as PhrS-mediated stimulation of *P. aeruginosa* quinolone signal and oxygen availability (line 315-317).

4. *The authors need to carry out additional experiments to elucidate the mechanism by which PhrS causes antitermination (by inhibiting Rho loading or translocation or by yet another mechanism).*

Response: Thank you for the deep insight. As suggested, we have now performed additional experiments to evaluate the mechanism of PhrS-mediated antitermination by inhibiting Rho loading or translocation mechanism. We performed crosslinked RNA immunoprecipitation (RIP) via His-Rho protein to investigate whether PhrS affects Rho loading or translocation with the CRISPR2 leader. We detected the gradually decreased enrichment of CRISPR2 leader in an increasing dose-dependent manner of PhrS (Fig. 4i), indicating that PhrS affects Rho loading. Consequently, to confirm PhrS-mediated inhibition of the biochemical interaction of CRISPR2 leader with Rho, RNA pull-down experiments were performed. Biotin-labeled CRISPR2 leader was transcribed in vitro, immobilized on streptavidin magnetic beads, hybridized to PhrS and incubated with His-Rho protein. Complexes captured on the beads were subjected to western blotting to identify and confirm that Rho binding gradually reduced due to the increased PhrS (Fig. 4j). Altogether, PhrS inhibits Rho loading to stimulate CRISPR2 transcription (line 210-219).

5. *The data presented in this study do not "reveal that bacterial intracellular signaling modifies the sequences of RNA molecules..." as stated on Page 11.*

Response: We apologize for the inappropriate description. We revised this sentence to “our study reveals an sRNA-mediated control of CRISPR-Cas system via RNA-RNA interaction, indicating that other RNA modifications at CRISPR loci, such as epigenetic modifications (M⁵C or M⁶A), may also play a part in the transcriptional or post-transcriptional control in the CRISPR-Cas adaptive immunity system, which requires further investigation.”.

6. *Figures need to be improved or commented on as follows:*

--*General: The choice of colors in the figures is distracting and confusing. There are too many different colors and some do not match the descriptions.*

Response: We have corrected them as suggested. We reduced the colors in the Figures, such as Fig. 2b, Fig. 2e, Fig. 2f, Fig.3d, Fig. 5a, and Fig. 5b. In addition, we also corrected the mismatches in descriptions about the colors or contents, such as Fig. 1c, Fig. 1d, etc.

--*Figure 1c and 1d: Pant391 and Pant463 should be labeled.*

Response: We have labeled them as suggested.

--Figure 1f: All bars should be labeled. It is unclear why these specific sRNAs were tested with the lacZ constructs. The authors should test all the positive hits with the lacZ clones, together with a few negative controls.

Response: We have added the full sRNA names to each lane in the Figure 1f.

Through a 274 sRNAs library screening, we identified 9 and 25 sRNA-CRISPR leader chimeras for CRISPR1 and CRISPR2 leader, respectively, by T4 RNA ligase-based assay (Fig. 1c,d, Supplementary Fig. 1b, and Supplementary Table 1). To investigate and characterize these 34 sRNAs regulating CRISPR loci in *P. aeruginosa*, we constructed each of the sRNA over-expressing plasmids in combination with *CRISPR1-lacZ* or *CRISPR2-lacZ* fusion plasmid and transformed them into PA14 to monitor lacZ activity. Of the 35 sRNAs tested, one sRNA pant391 repressed CRISPR2-lacZ expression by more than two-fold, while sRNAs pant463 and *PhrS* increased CRISPR2-lacZ expression (Fig. 1f). From these data, we would know which of these sRNAs clearly participate in CRISPR-Cas functionality. We have re-written this section (line 91-94). Moreover, we have added the data of an empty vector as a negative control shown in the new Fig. 1f.

--Figure 1g and Lines 88-89: It is not clear what is shown in this figure. What is meant by “unique amplicons”?

Response: We highly appreciate the careful reading and insightful comments. We have now reorganized the sentence to illuminate this figure: “To detect specific ligation of candidate targets of PhrS with CRISPR loci, we performed RT-PCR to analyze the ligated products as described in Fig. 1a using PhrS-specific primers and CRISPR locus-specific primers, followed by induction expression of RNA for up to 20 min in the presence of T4 RNA ligase or an inactive T4 RNA ligase (t4K99N). We noticed that the amplicons of PhrS-CRISPR2 leader chimeras were induced to facilitate the expression of *PhrS* for up to 20 min, but abrogated by an inactive T4 RNA ligase (Fig. 1g)”. The unique amplicons represent the *PhrS*-CRISPR2 leader chimeras amplicons (line 98-101).

--Figure 2c and Lines 102-104: This image is of very poor quality and should be improved. The authors should describe the phage JBD25 and JBD18 and their differences in more detail in the text.

Response: Thanks for your excellent suggestions. As suggested, we have improved the quality of the images (see new Fig. 2c) and also improved the quality of the images for Fig. 2g and Fig. 3e. Meanwhile, we added more information about the difference between the phage JBD25 and JBD18 in the main text: “We also observed that CRISPR-sensitive phage JBD25, which targets a spacer in CRISPR1 locus, failed to replicate in PA14 WT, $\Delta phrS$ and $\Delta phrS/p-phrS$ (Fig. 2c and Supplementary Fig. 1a). Conversely, CRISPR-sensitive JBD18, which targets a spacer in CRISPR2 locus, was able to replicate in PA14 $\Delta phrS$,

but failed to replicate in WT and $\Delta phrS/p-phrS$ (Fig. 2c), suggesting that these two phages can be used to demonstrate the specific locus targeting”.

--Figure 2d: The authors should provide an explanation for why CRISPR1 crRNA levels increase in $\Delta phrS$.

Response: We speculate that CRISPR1 crRNA is over-responsive, leading to the increased level. From the data of Figure 2a to 2c, there is no difference of CRISPR1 locus transcription and CRISPR1 interference between PA14 WT and $\Delta phrS$, indicating that PhrS has no effect on CRISPR1 locus transcription. We apologize for the oversight and have repeated the experiment and provided a new image to replace the old one (Fig. 2d).

--Figure 3a: The authors need to disclose the evidence for the secondary structure of PhrS.

Response: Thanks for the wonderful suggestion. The secondary structure of PhrS was identified by Elisabeth Sonnleitner et al (Fig. 7A and Fig. S4)⁴. In the carefully revised manuscript, we have added the content about this information (line 139-140).

--Figure 3c: The authors should note that the levels of 5S are significantly lower in the third lane and/or repeat this northern.

Response: We have repeated this northern blot as shown in the new panel, Figure 3c.

7. Editorial comments:

--Throughout: The manuscript needs to be checked for typographical errors such as Page 4, Line 111 “Consistent with these, expression...”, Page 6, Line 160 “Likewise, plating assay showed...”, Page 9, Line 283 “The precise mechanism of crew element of PhrS via direct...”

Response: We have carefully proofread the entire manuscript several times to correct all errors. For example: we have revised “the Page 6, Line 160...” to “Plated PA14 $\Delta phrS$ transformed with plader^{CRISPR2}-GFP plasmid displayed much weaker fluorescence compared to that of WT”. Thanks for your excellent comments.

--Throughout: sRNAs are typically written with the first and last letter capitalized and no italics i.e. “PhrS”

Response: We highly appreciate the great suggestion. We have carefully edited the entire manuscript as suggested.

--Line 162: The authors need to define the “creg” and “cmut” sequences of PhrS. What does this nomenclature mean? Is this a known motif? Why was this region chosen for mutational analysis? It would be helpful to show the base-pairing region on CRISPR2 at this point in the text to drive the reasoning for mutating the creg sequence.

Response: Thanks so much for the constructive suggestions. The secondary structures of PhrS was identified by *Sonnleitner et al*, who found that the PhrS contained a highly conserved **region** (creg element) of 12 nucleotides (nucleotides 169 to 182 downstream of the transcriptional start of *phrS*)⁴. Because conserved regions in homologous sRNAs from different species can serve as a tool to find target recognition sites within a sRNA⁵ and computational analysis by IntaRNA tool showed potential interaction between PhrS, especially creg element in PhrS, and +491 to +502 segments of CRISPR2. These are the reasons why we chose the creg motif as the candidate for mutational analysis. The “cmut” represents the point mutations introduced into creg element. We have re-organized to make these clearer to readers in the revised main text (line 139-143 and 151-156).

Reviewer #3, CRISPR expert (Remarks to the Author):

Lin et al uncover a novel small RNA-mediated mechanism by which Type I CRISPR-Cas systems are regulated in P. aeruginosa. The small RNA PhrS is the main focus of the paper—to identify PhrS, the authors first used a screen that relies upon T4 RNA ligase to covalently ligate any small RNAs that might be bound (via base-pair interactions) with the CRISPR1 and 2 leader regions. RT-PCR was then used to amplify sRNA-CRISPR leader hybrids and identify the small RNA linked to the leader. Following this screen, PhrS was identified along with a few other sRNA candidates. The authors then used genetic analyses to convincingly demonstrate that PhrS promotes efficient expression of the CRISPR 2 locus (but not the CRISPR 1 locus) and is required for CRISPR2 interference against plasmid transformation and phage infection. Using in silico analyses coupled with site-directed mutagenesis, the critical region in PhrS required for its function was identified (a so-called creg motif), and the region in the CRISPR leader that likely binds this small RNA was also found. Next, a series of in vitro and in vivo experiments were conducted to demonstrate that PhrS promotes transcription of the CRISPR locus by helping RNA polymerase to bypass a rho-dependent termination signal that appears in between the promoter and the first direct repeat in the CRISPR locus. Finally, this mode of regulation was shown to be present in other Type I CRISPR-Cas systems found in other strains of Pseudomonas.

Overall, this study constitutes a substantial amount of work that convincingly demonstrates the precise mechanism by which PhrS regulates the function of Type I CRISPR-Cas systems in P. aeruginosa. The flow of logic is very clear, and the authors have anticipated and answered many of my questions. I expect

that this work will be well-received by the CRISPR community because little is known about how CRISPR systems are regulated, and to my knowledge, this constitutes the very first report of a small RNA-mediated mechanism of CRISPR regulation. I have minor suggestions, most of which will help improve clarity and readability. They are as follows in no particular order:

1. To maintain high standards of rigor and reproducibility, it would be important to note the number of replicates that were performed for each experiment, either in the figure legends or materials and methods section.

Response: Thank you for this excellent suggestion. According to the advice, we have added the information about replicates to figure legends and methods section.

*2. Early in the results section (around lines 74-75), there should be a brief description of the architecture of the CRISPR loci in *P. aeruginosa* and a reference to Figure S1A.*

Response: Thanks for the wonderful suggestion. We added the information “*P. aeruginosa* PA14 I-F CRISPR-Cas system contains six *cas* genes and two CRISPR loci (Supplementary Fig.1a).” to describe the architecture of type I-F CRISPR-Cas system in *P. aeruginosa* PA14.

3. The difference between Figure 1d and 1e is very striking--there is almost no overlap with predicted interacting sRNAs and actual ones detected by the screen. Given that this screen can only detect interactions with sRNAs that have a 5'-monophosphate, while the majority of bacterial sRNAs have 5'-triphosphates, it would be important to indicate that the screen was in fact biased, and remove the word “unbiased” from the Discussion (line 243).

Response: We have rewritten “Through unbiased screening,” to “Through a high-throughput approach.”. Thanks for your excellent comments.

4. Somewhere around lines 102-105, it is important to explain in the text which CRISPR locus has a spacer that targets each phage, and refer to Fig. S1A. Without getting this information in the main text, it is impossible for the reader to interpret the phage plating data in Figure 2.

Response: We highly appreciate your very careful reading and great suggestion. We have added more information about this in the main text (line 116-1119): “We also observed that CRISPR-sensitive phage JBD25, which targets a spacer in CRISPR1 locus, failed to replicate in PA14 WT, $\Delta phrS$ and $\Delta phrS/p-phrS$ (Fig. 2c and Supplementary Fig.1a). Conversely, CRISPR-sensitive JBD18, which targets a spacer in CRISPR2 locus, was able to replicate in PA14 $\Delta phrS$, but failed to replicate in WT and $\Delta phrS/p-phrS$ (Fig. 2c).”

5. In lines 118-120, you should explain how you excluded possibility of the internal ORF having an effect. I know that the data appears in the supplementary figures but a brief explanation of this data should be included in the text.

Response: We apologize for the oversight. We revised this to “In addition to its regulatory function, PhrS has an ORF that encodes a highly conserved 37 amino-acid polypeptide (Supplementary Fig. 2a)⁶. We found that there is no difference of CRISPR2 locus transcription between PA14 $\Delta phrS$ and $\Delta phrS/phrS-ORF$ (restored expression of internal ORF of PhrS) by *lacZ* reporter and northern blotting (Supplementary Fig. 2b,c) and similarly no difference of CRISPR-Cas interference was noted (Supplementary Fig. 2d,e). These data demonstrate that the internal ORF of PhrS-encoded protein had no effect on CRISPR-Cas functionality, indicating that PhrS as a sRNA may act on PA14 CRISPR-Cas adaptive immunity” (line 133-136).

6. As above, lines 133-136 rule out some possible mechanisms of action of PhrS and simply refer the reader to Figures S3 and S4. It would be important to give a brief explanation of what the data is showing so that the reader can follow.

Response: Thank you for this excellent suggestion. We added the following explanation: “However, mutation of PqsA-E has no effect on CRISPR2 locus transcription (Supplementary Fig. 3a,b) and consequent CRISPR-Cas interference (Supplementary Fig. 3c,d), indicating that *PhrS*-mediated PQS biosynthesis operon had no role in CRISPR-Cas expression or function. As several sRNA molecules regulate gene function by directly binding to their target chromosomal DNAs⁷, we used a ‘reverse transcription-associated trap (RAT)’ assay⁸ to detect RNA/DNA interaction through pull-down and PCR by interacting DNA-specific primers (Supplementary Fig. 4a). We observed that no interaction between PhrS and CRISPR2 locus chromatin DNAs as well as *casI* locus and *pheS* locus (Supplementary Fig. 4b)” (line 166-173).

7. Regarding Figure S4, what is the positive control? Also, it seems strange that one step of the procedure involves “nuclei collection” when bacteria do not have nuclei. Please revise the description of the protocol to be more accurate.

Response: We highly appreciate your great comments and have revised the description about this information to “bacteria chromosomal DNA collection”. The positive control is *IRAIN* lncRNA binding to chromatin DNAs that we refer to the method of “RAT” assay⁸. We have added more description about this in the supplementary Fig. 4b legend.

8. Line 135—please change “chromatin” to “chromosomal” since bacteria do not have chromatin.

Response: We have corrected it as suggested.

9. Lines 155-156—please fix the grammar in that sentence.

Response: Thanks for your advice. We have revised this to “To test whether PhrS influences Rho activity at CRISPR2 leader and performed a GFP assay in PA14 WT and $\Delta phrS$ containing pleader^{CRISPR2}-GFP plasmid (Fig. 4a).”

10. Line 223—what is meant by “constituted”? Please use a better descriptive word.

Response: We have revised “we constituted two other types of CRISPR-Cas systems...” into “we used two other types of CRISPR-Cas systems...”

11. Line 224—Please change “another” to “other”.

Response: We have corrected it as suggested.

12. Lines 293-297—The data does not appear to support the claim that the data shows “...bacterial intracellular signaling modifies the sequences of RNA molecules to control CRISPR-Cas systems...” Please revise to more accurately reflect the findings.

Response: We apologize for the unclear description. We revised this sentence to “our study reveals an sRNA-mediated control of CRISPR-Cas system via RNA-RNA interaction, indicating that other RNA modifications at CRISPR loci, such as epigenetic modifications (M⁵C or M⁶A), may also play a part in the transcriptional or post-transcriptional control in the CRISPR-Cas adaptive immunity system, which requires further investigation.”

13. In Fig. 1F, please label the small RNAs that each bar corresponds to. Also, Pant 463 and 391 do not appear to be in the annotated list. Are they among the non-annotated ones?

Response: Thanks for your suggestion. We labeled each sRNA in its respective lane. The pant463 and pant391 are non-annotated sRNA at present. We hope to define their functionality in the CRISPR-Cas system in future.

14. In Fig. 5A, please change “pro-crRNA” to “pre-crRNA”.

Response: We have revised the “pro-crRNA” to “pre-crRNA” in the Fig. 5A

15. In Fig. 6B, please change “Phrs” to “PhrS”.

Response: We have revised it.

Reference

- 1 Han, K., Tjaden, B. & Lory, S. GRIL-seq provides a method for identifying direct targets of bacterial small regulatory RNA by in vivo proximity ligation. *Nat Microbiol* **2**, 16239, doi:10.1038/nmicrobiol.2016.239 (2016).
- 2 Gómez-Lozano, M., Marvig, R. L., Molin, S. & Long, K. S. Genome-wide identification of novel small RNAs in *Pseudomonas aeruginosa*. *Environmental microbiology* **14**, 2006-2016 (2012).
- 3 Lu, P., Wang, Y., Hu, Y. & Chen, S. RgsA, an RpoS-dependent sRNA, negatively regulates rpoS expression in *Pseudomonas aeruginosa*. *Microbiology* **164**, 716-724 (2018).
- 4 Sonnleitner, E. *et al.* The small RNA PhrS stimulates synthesis of the *Pseudomonas aeruginosa* quinolone signal. *Mol Microbiol* **80**, 868-885, doi:10.1111/j.1365-2958.2011.07620.x (2011).
- 5 Sharma, C. M., Darfeuille, F., Plantinga, T. H. & Vogel, J. A small RNA regulates multiple ABC transporter mRNAs by targeting C/A-rich elements inside and upstream of ribosome-binding sites. *Genes Dev* **21**, 2804-2817, doi:10.1101/gad.447207 (2007).
- 6 Sonnleitner, E. *et al.* Detection of small RNAs in *Pseudomonas aeruginosa* by RNomics and structure-based bioinformatic tools. *Microbiology* **154**, 3175-3187, doi:10.1099/mic.0.2008/019703-0 (2008).
- 7 Waters, L. S. & Storz, G. Regulatory RNAs in bacteria. *Cell* **136**, 615-628, doi:10.1016/j.cell.2009.01.043 (2009).
- 8 Sun, J. *et al.* A novel antisense long noncoding RNA within the IGF1R gene locus is imprinted in hematopoietic malignancies. *Nucleic Acids Res* **42**, 9588-9601, doi:10.1093/nar/gku549 (2014).

REVIEWERS' COMMENTS:

Reviewer #1 (Remarks to the Author):

The authors have made a commendable effort to address the reviewers' comments. I think there will be substantial interest in the findings.

Reviewer #3 (Remarks to the Author):

The authors appear to have addressed all of my initial concerns.

Point-by-point responses to reviewers' comments:

REVIEWERS' COMMENTS:

Reviewer #1 (Remarks to the Author):

The authors have made a commendable effort to address the reviewers' comments. I think there will be substantial interest in the findings.

Response: We extremely appreciate your great suggestion to improve our manuscript.

Reviewer #3 (Remarks to the Author):

The authors appear to have addressed all of my initial concerns.

Response: We highly appreciate your careful reading and the great suggestion to improve our manuscript.